1 **Further improvement of wet process treatments in GEOS-Chem v12.6.0: Impact on global**

2 **distributions of aerosols and aerosol precursors**

4 Gan Luo[1], Fangqun Yu[1], Jonathan M. Moch[2]

6 [1]Atmospheric Sciences Research Center, University at Albany, Albany, NY, USA

7 [2]Department of Earth and Planetary Sciences, Harvard University, Cambridge, MA, USA

**Abstract**

Wet processes, including aqueous phase chemistry, wet scavenging, and wet surface

uptake during dry deposition, are important for global modeling of aerosols and aerosol

precursors. In this study, we improve the treatments of these wet processes in the GEOS-Chem

v12.6.0, including pH calculations for cloud, rain, and wet surfaces, the fraction of cloud

available for aqueous phase chemistry, rainout efficiencies for various types of clouds, empirical

washout by rain and snow, and wet surface uptake during dry deposition. We compare simulated

surface mass concentrations of aerosols and aerosol precursors with surface monitoring networks

over the United States, Europe, Asia, and Arctic regions, and show that model results with

updated wet processes agree better with measurements for most species. With the

implementation of these updates, normalized mean biases (NMB) of surface nitric acid, nitrate,

and ammonium are reduced from 78%, 126%, and 45% to 0.9%, 15%, and 4.1% over the US

sites, from 107%, 127%, and 90% to -0.7%, 4.2%, and 16% over Europe sites, and from 121%,

269%, and 167% to -21%, 37%, and 86% over Asia remote region sites. Comparison with

surface measured $SO_2$, sulfate and black carbon at four Arctic sites indicated that these species

simulated with the updated wet processes match well with observations except for a large

underestimate of black carbon at one of the sites. We also compare our model simulation with

aircraft measurement of nitric acid and aerosols during the ATom-1 and ATom-2 periods and

found a significant improvement of modeling skill of nitric acid, sulfate, and ammonium in the

Northern Hemisphere during winter time. The NMBs of these species are reduced from 163%,

78%, and 217% to -13%, -1%, and 10%, respectively. The investigation of impacts of updated

wet process treatments on surface mass concentrations indicated that the updated wet processes

have strong impacts on the global means of nitric acid, sulfate, nitrate, and ammonium and

relative small impacts on the global means of sulfur dioxide, dust, sea salt, black carbon, and organic carbon.

## 1. Introduction

Aqueous phase chemistry, wet scavenging, and wet surface uptake during dry deposition are the three major atmospheric wet processes for aerosols and aerosol precursors. Aqueous phase chemistry plays a role as reaction chamber which efficiently converts aerosol precursors to aerosols (Ervens et al.,2011; Walcek and Taylor,1986). Wet scavenging, a process by which chemicals accumulate in droplets and then are removed by precipitation, is the predominant removal pathway of aerosols and aerosol precursors (Textor et al., 2006). Dry deposition, where chemicals settle out of the atmosphere in the absence of precipitation, is greatly enhanced due to the absorption of water soluble gases at wet surfaces associated with dew, fog, and rain (Garland and Branson, 1977; Wesely, 1989). These wet processes significantly impact global mass load and redistribute aerosols and aerosol precursors. Aerosol mass load and its global distributions are important for studies of aerosol optical properties (Kinne et al., 2006), aerosol direct radiative forcing (Myhre et al., 2013; Penner et al., 1994), and the health effects of particulate matter (Shiraiwa et al., 2017; Hopke et al., 2006). A better representation of wet processes in global modeling of aerosols and aerosol precursors can therefore enhance our ability to accurately simulate these different aerosol impacts.

GEOS-Chem is a widely used community model which is continuously being improved (Holmes et al., 2019; Keller et al., 2014; Martin et al., 2003; Bey et al., 2001). Luo et al. (2019), L2019 hereafter, updated the GEOS-Chem wet scavenging scheme by using the Modern-Era Retrospective analysis for Research and Applications, Version 2 (MERRA-2) spatially and temporally varying cloud and rain water to replace the assumption of fixed in-cloud condensation water (ICCW) in the GEOS-Chem rainout parameterization and by using new empirical rates for nitric acid and water soluble aerosols in washout. These changes together reduced the normalized mean biases (NMB) of simulated nitric acid, nitrate, and ammonium mass concentrations at the United States' surface monitoring networks from 145%, 168%, and 81% to 24%, 25%, and 13%, respectively. However, the impacts of the updated wet scavenging scheme on simulations over other regions (Europe, Asia, and remote areas) and free troposphere were not investigated. Moreover, L2019 only investigated the changes of nitric acid, nitrate, and

ammonium. The impact of the updated wet scavenging scheme on other aerosols such as sulfate, sea salt, dust, and carbonaceous aerosols was not investigated in that work. Due to the large impact of updated wet scavenging on model simulations, a comprehensive validation of simulated aerosols and aerosol precursors with ground based monitoring networks for surface mass concentrations and aircraft measurements for vertical profiles is needed.

In this study, we further update the treatments of wet processes (aqueous chemistry, wet scavenging, and wet surface uptake during dry deposition) in GEOS-Chem and evaluate comprehensively simulated major inorganic aerosol precursors (sulfur dioxide, nitric acid, and ammonia) and aerosols (sulfate, nitrate, ammonium, black carbon, and organic carbon) by comparison with a large set of in-situ observations. The updates to the wet processes are detailed in Section 2. Comparisons of simulations with measurements from surface monitoring networks including the United States Environmental Protection Agency (USEPA), the Interagency Monitoring of Protected Visual Environments (IMPROVE), the Chemical Speciation Network (CSN), the Clean Air Status and Trends Network (CASTNET), the Ammonia Monitoring Network (AMoN), the National Trends Network (NTN), the European Monitoring and Evaluation Programme (EMEP), and the Acid Deposition Monitoring Network in East Asia (EANET) are given in section 3.1. Validations of aerosols and aerosol precursors for the Arctic and the Atmospheric Tomography (ATom) mission are presented in sections 3.2 and 3.3. The impact of the updated wet processes on global surface concentrations or aerosols and aerosol precursors are discussed in section 3.4. A summary of our results is given in section 4.

**2 Updates of wet process treatments in GEOS-Chem associated with aerosol precursor and aerosol modeling**

In the publicly released GEOS-Chem version 12.6.0, GC12 thereafter, in-cloud aqueous phase chemistry was developed by Chin et al. (2000) for $SO_2$. The wet scavenging scheme, including rainout due to formation of precipitation from clouds and washout due to falling precipitation from upper layers, was developed by Jacob et al. (2000) and Liu et al. (2001) for aerosols and by Amos et al. (2012) for gases. Scavenging of aerosol by snow and cold–mixed precipitation was updated by Wang et al. (2011, 2014). Wet surface uptake during dry deposition is represented with constant values of effective Henry's law coefficient for surface resistance

calculations (http://wiki.seas.harvard.edu/geos-chem/index.php/Physical_properties_of_GEOS-

Chem_species#Definition_of_Henry.27s_law_constants).

L2019 showed that the assumption of in-cloud condensation water with a fixed value (1

g·m$^{-3}$) in the rainout parameterization in GC12 is one of the major reasons causing an

overestimate in nitrate and ammonium mass concentrations compared to surface monitoring

networks over the US. After replacing the fixed value of in-cloud condensation water with

MERRA-2 cloud and rain water, we get an updated equation for rainout loss fraction (Luo et al.,

2019):

$$F = \frac{P_{\mathrm{r}}}{k \cdot \mathrm{ICCW}}\left(1 - e^{-k \cdot \Delta t}\right) = \frac{f_{\mathrm{c}} \cdot P_{\mathrm{r}}}{k\left(\mathrm{LCW} + \mathrm{ICW} + P_{\mathrm{r}} \cdot \Delta t\right)}\left(1 - e^{-k \cdot \Delta t}\right), \quad (1)$$

where $F$ is the fraction of a water-soluble tracer in the grid-box scavenged by rainout, $\Delta t$ (s) is

the model integration time step. $k$ is the first-order rainout loss rate which represents the

conversion of cloud water to precipitation water. ICCW (g·m$^{-3}$) is in-cloud condensation water.

$P_{\mathrm{r}}$ (g·m$^{-3}$·s$^{-1}$) is the rate of new precipitation formation. $f_{\mathrm{c}}$, LCW (g·m$^{-3}$), and ICW (g·m$^{-3}$) are the

grid-box mean cloud fraction, , liquid phase cloud water content, and ice phase cloud water

content, respectively.

L2019 also showed that the difference between observations and simulations can be

further reduced, through (1) the update of empirical washout coefficients by rain for water-

soluble aerosol with the value which was calculated by the parameterization of Laakso et al.

(2003) for a 500 nm particle diameter, and (2) the new estimated washout coefficients for nitric

acid by referring to field measurements for particles with a 10 nm diameter (Laakso et al., 2003)

and the theoretical dependence of scavenging coefficients on particle sizes for particles < 10 nm

(Henzing et al., 2006). L2019 only focused on warm cloud wet scavenging, and did not

systematically consider the impact of wet process treatments on the simulated aerosols and

aerosol precursors. Here we show that a number of treatments in GC12 and L2019 can be further

updated (as detailed below) to improve the performance of GEOS-Chem in simulating spatial

and temporal variations of major aerosols and aerosol precursors on a global scale.

**2.1 pH for cloud, rain, and wet surface**

Water pH is important for dissolution and subsequent aqueous phase reactions of water-

soluble gases (Turnock et al., 2019; Ervens, 2015; Pandis and Seinfeld, 1989). Based on Henry's

law, dissolution of water-soluble gases can be calculated as:

$$f_w = 1 - \frac{1}{1 + H^* \cdot R \cdot T \cdot LW}, \quad (2)$$

where $f_w$ is the dissolution fraction for water-soluble gases, $H^*$ (mol·L⁻¹·atm⁻¹) is effective

Henry's law constant, $R$ (0.08205 L·atm·K⁻¹·mol⁻¹) is the gas constant, $T$ (K) is the temperature,

and LW (m³·m⁻³) is the liquid water content.

$H^*$ represents the impact of temperature, water acidity, and aqueous phase equilibrium on

solubility of water-soluble species (Seinfeld and Pandis, 2016). For $SO_2$, $H_2O_2$, and $NH_3$, which

are important for aerosol precursor and aerosol simulation, $H^*$ can be calculated as (Seinfeld and

Pandis, 2016):

$$\left.\begin{array}{l} H^*_{SO2} = H_{SO2}\left(1 + \dfrac{K_1}{[H^+]} + \dfrac{K_1 \cdot K_2}{[H^+]^2}\right), \\[2ex] H_{SO2} = 1.22 e^{10.55\left(\frac{298.15}{T}-1\right)}, \\[2ex] K_1 = 1.3 \times 10^{-2} e^{6.75\left(\frac{298.15}{T}-1\right)}, \\[2ex] K_2 = 6.31 \times 10^{-8} e^{5.05\left(\frac{298.15}{T}-1\right)} \end{array}\right\} \quad (3)$$

$$\left.\begin{array}{l} H^*_{H2O2} = H_{HO2}\left(1 + \dfrac{K_3}{[H^+]}\right), \\[2ex] H_{H2O2} = 8.3 \times 10^4 e^{24.82\left(\frac{298.15}{T}-1\right)}, \\[2ex] K_3 = 2.2 \times 10^{-12} e^{12.52\left(\frac{298.15}{T}-1\right)} \end{array}\right\} \quad (4)$$

$$\left.\begin{array}{l} H^*_{NH3} = H_{NH3}\left(1 + \dfrac{K_5[H^+]}{K_4}\right), \\[2ex] H_{NH3} = 59.8 e^{14.1\left(\frac{298.15}{T}-1\right)}, \\[2ex] K_4 = 1. \times 10^{-14} e^{-22.5\left(\frac{298.15}{T}-1\right)}, \\[2ex] K_5 = 1.7 \times 10^{-5} e^{-14.5\left(\frac{298.15}{T}-1\right)} \end{array}\right\} \quad (5)$$

where $H_{SO2}$, $H_{H2O2}$, and $H_{NH3}$ are the Henry's law constants (M atm$^{-1}$) for $SO_2$, $H_2O_2$, and $NH_3$, respectively. $K_1$ (M), $K_2$ (M), $K_3$ (M), $K_4$ (M$^2$), and $K_5$ (M) are rate coefficients for $SO_2$ reaction, $HSO_3^-$ reaction, $H_2O_2$ reaction, $H_2O$ reaction, and $NH_3$ reaction, respectively. The values of the Henry's law constants and rate coefficients are the same as those used in GEOS-Chem aqueous phase chemistry. [H$^+$] (M) is the hydrogen ion concentration in cloud/rain droplets and at wet surfaces, which is related to pH as:

$$[H^+]=10^{-pH}, (6)$$

GC12 calculates cloud water pH iteratively by using the concentrations of sulfate, total ammonium (ammonium + ammonia), total nitrate (nitrate + nitric acid), $SO_2$, and $CO_2$ based on their effective Henry's law coefficients and cloud liquid water content in corresponding grid box (Alexander et al., 2012). This iterative calculation is updated to use Newton's method in order to arrive at a consistent result (Moch et al., 2020). To implement Newton's method the equilibrium expressions for the concentrations of each soluble semi-volatile ion (SSVI) in terms of H+ and the derivatives for these equilibrium expressions are each solved explicitly so that the Newton's method equation is in the form of:

$$H_{n+1}^+ = H_n^+ + \frac{\left[SSVI\left(H_n^+\right)\right] + [SNVI]}{\dfrac{d}{dH^+}\left[SSVI\left(H_n^+\right)\right]}, (7)$$

where SNVI is the concentrations of soluble nonvolatile ions. For equation (7) the concentrations of each ion are multiplied by the ion charge (e.g. the terms for $SO_3^{2-}$ concentrations are multiplied by -2).

In tests with this new calculation the solution always converged to an answer in less than 20 iterations, but if a maximum of 50 iterations is reached we set it so that the last two solutions are averaged together. We here considered the solution to converge if the difference between $H_n^+$ and $H_{n+1}^+$ was less than 0.01. By default the initial guess for H$^+$ is set to 4.5, but we tested initial guesses ranging from a pH of 2 to 13 and found no change in the values at which the answer converged.

To represent the removal of aerosols due to rainout, GC12 assumes 30% of sulfate, nitrate, and ammonium are removed away from cloud water before cloud water pH calculation. To take into account the variations in the amount of these species rained out, we use the real-time

rainout fractions for corresponding species which are calculated during the treatment of wet

scavenging to replace this constant value (i.e., 30%). Additionally, in GC12, sulfate is assumed

to be the only SNVI in cloud water, while ammonium and nitrate are treated as volatile species

similar to ammonia and nitric acid:

$$[\text{SNVI}] = 2\left[\text{SO}_4^{2-}\right], \quad (8)$$

Previous studies found that observed ammonium-sulfate aerosol molar ratio is lower than

2 over the US (Silvern et al., 2017; Hidy et al., 2014). Guo et al. (2018) found ammonium-sulfate

aerosol molar ratio during the Wintertime Investigation of Transport, Emissions, and Reactivity

(WINTER) study to be 1.47±0.43 and pointed out that this phenomena indicates an important

role of soluble nonvolatile cations in aerosol thermodynamics. To reflect the impact of soluble

nonvolatile cations on cloud water pH, we assume that total amount of soluble nonvolatile

cations associated with aerosol thermodynamics (SNVC) is 25% of sulfate. We also consider the

contribution of calcium and magnesium based on simulated dust mass in GC12, assuming that 3%

of dust mass is soluble calcium and 0.6% is soluble magnesium (Farlie et al., 2010; Moch et al.,

2020) , to soluble nonvolatile ions (SNVI):

$$[\text{SNVI}] = 2\left[\text{SO}_4^{2-}\right] - 2[\text{SNVC}] - 2\left[\text{Ca}^{2+}\right] - 2\left[\text{Mg}^{2+}\right], \quad (9)$$

Rainwater pH, which is used for the calculation of effective Henry's law constants of

water-soluble gases in rain droplets (Eqs. 3-5), is assumed to be a constant value of 4.5 in GC12.

Rainwater pH is determined by the cloud water pH where the rain is produced, uptake of water

and ions during rainfall processes, and evaporation of rain droplets. In addition, rainwater pH

also depends on temperature (Smith and Martell, 1976). Although it is difficult to fully trace

rainwater pH in the model based on current available information in GC12, we use cloud pH at

where rainout occurs to represent rainwater pH for rainout process and rainwater-mass-weighted

cloud pH above where washout occurs to represent rainfall water pH for washout process in this

work. The calculated rainwater pH in this study varied from 4.3 to 6.9.

pH values also affect dry deposition of water-soluble gases via its impact on the uptake

due to dissolution at wet surfaces. The origin of surface water where this uptake occurs is

therefore important to account for the effect if varying pH. GC12 calculated effective Henry's

constant for dry deposition by assuming temperature of 298.15 K and leaf water pH of 7. Surface

water on land is dominated by leaf water whose pH is ~7. The pH of ocean surface water varies

from 8 to 8.5 (Antonov, 2010; Jacobson, 2005). de Caritat et al. (2005) found the pH of the meltwaters of the Arctic snow varies from 4.6 to 6.1 with median value of 5.4. We assume the pH values at wet surface are 7 for land, 8.2 for ocean, and 5.4 for snow in this work.

## 2.2 Fraction of cloud available for aqueous phase chemistry

In GC12, the fraction of cloud available for aqueous phase chemistry is assumed to be 100% of grid box cloud fraction when temperatures are above 258 K and 0% of grid box cloud fraction when temperatures are below 258 K. This means aqueous phase chemistry in mixed clouds where temperatures are often below 258 K is not considered in GC12. However, many studies have indicated that supercooled cloud water can exist when temperatures are above 237 K (Rosenfeld and Woodley, 2000; Sassen, 1985). Therefore, we calculate aqueous phase cloud fraction based on MERRA-2 cloud liquid content and cloud ice content when temperatures are higher than 237 K:

$$f_{aq} = f_c \frac{\text{LCW}}{\text{LCW+ICW}}, \ (T > 237 \text{ K}) \ , (10)$$

where $f_{aq}$ is aqueous phase cloud fraction, LCW (g m$^{-3}$) is grid box mean liquid phase cloud water content, and ICW (g m$^{-3}$) is grid box mean ice phase cloud water content.

## 2.3 Rainout efficiencies

### 2.3.1 Warm cloud

GEOS-Chem uses rainout efficiencies to represent the absorptions of water-soluble gasses and aerosols in the cloud condensate phase (Jacob et al., 2000; Mari et al., 2000; Liu et al., 2001). After applying these efficiencies with the updated parameterization for rainout loss fraction (Luo et al., 2019), we get the new equation as

$$F = \frac{f_c \cdot P_r}{k \left( \text{LCW+ICW} + P_r \cdot \Delta t \right)} \left( 1 - e^{-E_r \cdot k \cdot \Delta t} \right), (11)$$

where $E_r$ is the rainout efficiency for corresponding species. Eq. (11) is the same as Eq. (1) except Eq. (11) contains $E_r$ in the rainout calculation.

In GC12, rainout efficiencies for water-soluble aerosols are assumed to be 100% while those for water-soluble gases, except nitric acid and SO$_2$, are calculated via Henry's law constants (Jacob et al., 2000). $E_r$ of nitric acid is assumed to be the same as water-soluble

aerosols due to its high solubility. $E_r$ of $SO_2$ is assumed to be the same as water-soluble aerosols but limited by the availability of $H_2O_2$ in the precipitating grid box (Chin et al., 1996). It means rainout of $SO_2$ in GC12 is attribute to the aqueous phase oxidation of $SO_2$ by $H_2O_2$ rather than the absorption by cloud water. However, GEOS-Chem already accounted for in-cloud oxidation of $SO_2$ as part of the aqueous phase chemical calculation which converts in-cloud $SO_2$ to sulfate, so doing the same in the scavenging calculation would be double-counting the removal of $SO_2$. Considering the low solubility of $SO_2$ in water, it is more appropriate to calculate rainout efficiency for $SO_2$ based on Henry's law. In the present work, we assume $E_r$ of $SO_2$ equals its dissolution fraction:

$$E_{r\_SO2} = f_{w\_SO2} , (12)$$

with $f_{w\_SO2}$ calculated with Eq. (2).

In the present work, we also modified rainout efficiencies for hydrophilic black carbon (BC) and primary organic carbon (POC), from 100% in GC12 to 50%. The rationale for the modification is that, although the aging of BC and POC in the atmosphere converts these aerosols from hydrophobic to hydrophilic, they are not as easily activated into cloud droplet as water-soluble aerosols (e.g. sulfate, nitrate, ammonium). The composition of the particles decides the hygroscopic parameter kappa which is important for cloud activation calculation (Abdul-Razzak et al., 2000). If BC and POC are internally mixed with the sulfate, nitrate, ammonium (SNA) aerosols, then they all have similar compositions. However, in the actual atmosphere, many particles are externally mixed: some particles are pure SNA while others are primary particles (BC, POC, dust, etc.) coated with SNA (Fassi-Fihri et al., 1997). It takes time for primary particles to gain coating through condensation, coagulation, and aqueous chemistry. The amount of SNA coated on primary particles depends on the aging time and abundance of SNA in the air. Based on a detailed size and mixing state resolved advanced particle microphysics (APM) simulation which explicitly resolves the amount of SNA coating (Yu et al., 2012), the hygroscopic parameter kappa of coated BC and POC is roughly about half of that of SNA. More robust calculation of rainout efficiencies for BC and POC should consider the amount of soluble species coated on these particles (Yu et al., 2012; Yu and Luo, 2009), but this will be the subject of future work.

**2.3.2 Mixed and cold clouds**

In GC12, aerosols in mixed clouds (237 K ≤ T $<$ 258 K) and cold clouds (T < 237 K) were assumed to be removed through heterogeneous and homogeneous freezing nucleation (Wang et al., 2014). GEOS-Chem assumed that heterogeneous nucleation dominates ice formation at 237 K ≤ T $<$ 258 K (mixed cloud) and results in 100% rainout efficiencies only for dust and hydrophobic black carbon which are considered as ice nuclei (IN). Homogeneous nucleation takes place at T < 237 K (cold cloud) and results in 100% rainout efficiencies for both water-soluble aerosol and IN.

Ice nucleation processes and their impacts on aerosol wet scavenging by mixed and cold clouds are largely unclear. However, it is known that ice nucleation rates depend strongly on temperature (DeMott et al., 2015; Kanji and Abbatt, 2010). To take into account this, we parameterize rainout efficiencies at warmer temperatures based on the fraction of dust in mixed clouds contributing to IN, which can be calculated as a function of T according to DeMott et al. (2015) as:

$$E_{r\_mixed\_dust} = \frac{e^{0.46(273.16-T)-11.6}}{153.5}, \quad (237\ K \leq T < 258\ K) \ , \ (13)$$

In addition to T, ice nucleation efficiency of particles also depends on their sizes and smaller particles (diameter $<$ 500 nm) are less likely to act as IN (Niedermeier et al., 2015). While most of the mass of dust particles are dominated by those larger than 500 nm (Zender et al., 2003), a significant fraction of BC particles are smaller than 500 nm (Sahu et al., 2012). Based on sectional aerosol microphysics calculation in GEOS-Chem-APM (Yu and Luo, 2009), the mass fraction of BC particles with diameter $>$ 500 nm is ~50%. In this study, we assume $E_r$ for hydrophobic BC in both mixed cloud (237 K ≤ T $<$ 258 K) and cold cloud (T < 237 K) are 50% of those values for dust.

Water-soluble aerosols are 100% removed via homogeneous freezing nucleation in cold cloud (Wang et al., 2014; Liu et al., 2001). Strom et al. (1997) observed that ~ 40% of preexisting aerosol mass is incorporated in ice crystals. In this work, we assume cold cloud rainout efficiencies are 40% for water-soluble aerosol, 50% for hydrophobic black carbon, and 100% for dust, respectively. Additionally, rainout of cold clouds is limited to below the MERRA-2 troposphere since stratospheric water in MERRA-2 is known to have unphysical behavior.

In GC12, cold cloud wet scavenging of nitric acid is treated the same as water-soluble

aerosol. However, in cold clouds (T < 237 K), nitric acid is removed by the partitioning on ice

crystals (Kärcher and Voigt, 2006; Voigt et al., 2006), while water-soluble aerosol is removed by

homogeneous freezing nucleation. Kärcher et al. (2008) used a climatology of cirrus ice water

content together with observed molar ratios of $HNO_3/H_2O$ in cirrus ice particles to estimate the

range of nitric acid content in cirrus ice (185-240 K). Their study showed that less efficient nitric

acid uptake limits the nitric acid content in cirrus ice at higher temperatures and small ice water

contents permit only little nitric acid in ice at low temperatures. The fraction of nitric acid in ice

generally increases with decreasing temperature. Kärcher and Voigt (2006) attributed this

behavior to less efficient nitric acid trapping at higher temperatures despite faster ice growth

rates, which is caused by increasingly rapid escape of adsorbed nitric acid into the gas phase. A

parameterization of nitric acid partitioning in cold cloud developed by Kärcher et al. (2008) is

employed here to calculate $E_r$ of nitric acid in cold cloud when temperature is below 240 K:

$$E_r = \frac{10^{-(-26.5 \times 1.00155^T + 30.7)} \cdot \frac{63}{18} \cdot [\frac{LCW+ICW}{f_c}]_{vmr}}{[HNO_3]_{vmr}} \ , (14)$$

where $[\frac{LCW+ICW}{f_c}]_{vmr}$ is volume mixing ratio of in-cloud water and in-cloud ice, and

$[HNO_3]_{vmr}$ is volume mixing ration for nitric acid gas.

**2.4 Empirical washout coefficients by rain and snow**

Washout coefficients by rain and snow in GC12 were updated by Wang et al. (2011) by

adopting the parameterization constructed by Feng (2007, 2009) for individual aerosol modes.

Accumulation-mode washout coefficients were used for all aerosols except dust and sea salt, for

which the coarse mode coefficients were used. Previous studies noticed that washout rates by

rain derived from field measurements are 1 to 2 orders of magnitude larger than the values from

theoretical calculation (Wang et al., 2010; Luo et al., 2019). Therefore, L2019 recommended

using empirical washout coefficients for the simulation of washout by rain.

Wang et al. (2014) found that the large differences in washout rate between field

measurements and theoretical calculation not only appear in washout by rain but also appear in

washout by snow. In this work, we use the semi-empirical parameterization developed by Wang

et al. (2014) for the calculation of nitric acid and aerosol washout by both rain and snow.

Washout rate is calculated by an exponential equation:

$$k_{wash} = \Lambda \left( \frac{P_d}{f_r} \right)^b , (15)$$

where $k_{wash}$ (s$^{-1}$) is the washout rate, $P_d$ (mm h$^{-1}$) is rain or snow falling from upper layers, $f_r$ is

rainfall area fraction, $\Lambda$ is washout scavenging coefficient, and $b$ is an exponential coefficient.

The values of $\Lambda$ and $b$ for nitric acid and aerosol washout by rain (T > 268 K) and snow

(248 K < T < 268 K) are shown in Table 1. We assume precipitation at temperatures lower than

248 K is dominated by ice. GC12 assumed washout of aerosol by ice is the same as that by snow.

However, uptake of aerosol by ice and by snow is different. Schneider et al. (2019) found

specific surface area (SSA) of ice crystal is ~1/5 of SSA of snow. Therefore, in this work, we

roughly assume washout rate by ice (T < 248 K) is 1/5 of that by snow. Washout of nitric acid

uses the same values as in L2019 but we extend the temperature limitation from 268 K to 248 K.

Washout of nitric acid by ice is assumed to be 1/5 of that by snow. Empirical washout

coefficients by rain and snow for coarse aerosol and hydrophobic fine aerosol in this work are

based on the values in Wang et al. (2014). Because the rain washout rate for water-soluble

aerosols measured by Laakso et al. (2003) is still ~ 20 times larger than that calculated by the

semi-empirical parameterization, we used the value of $1\times10^{-5}$ to replace $5\times10^{-7}$ for hydrophilic

aerosol's washout by rain. The washout coefficient of hydrophilic aerosol by snow is replaced

with the value of $2\times10^{-4}$ which is 20 times higher than the value by rain. The assumption of

different washout coefficients for hydrophobic and hydrophilic aerosols is because the rain

washout rate for water-soluble aerosols measured by Laakso et al. (2003) is larger than that

calculated by the semi-empirical parameterization. One of the possible reasons is that droplet–

particle collection mechanisms for hydrophobic and hydrophilic aerosols are different. Washout

by ice is assumed to be 1/5 of that by snow.

**2.5 Wet surface uptake during dry deposition**

Uptake of water-soluble gases at wet surfaces is strongly influenced by dissolution

processes. The solubility of $SO_2$, $H_2O_2$, and $NH_3$ at wet surfaces needs to be calculated via

effective Henry's law coefficient because it is associated with a series of aqueous phase reactions

(Seinfeld and Pandis, 2016). In GC12, $H^*$ of $SO_2$, $H_2O_2$, and $NH_3$ for dry deposition are assumed

to be the constants with the values of $10^5$ M atm$^{-1}$, $5\times10^7$ M atm$^{-1}$, and $2\times10^4$ M atm$^{-1}$, respectively (http://wiki.seas.harvard.edu/geos-chem/index.php/Physical_properties_of_GEOS-Chem_species#Definition_of_Henry.27s_law_constants). In this work, we consider the impacts of temperature and pH at wet surface on the values of $H^*$ (Erisman et al., 1994; Wesely et al., 1990), and the values of $H^*$ for $SO_2$, $H_2O_2$, and $NH_3$ are calculated with equations (3-5). Wet surface pHs discussed in section 2.1 are used to reflect the impact of wet surface acidity on dissolution during dry deposition. Wet surface pHs are only determined by land type and are not altered by precipitation. Ganzeveld et al. (1998) reported that observations and physical-chemical model simulations indicated $SO_2$ dry deposition velocity increases from a minimum value of 0.01 cm s$^{-1}$ for a temperature of 253 K to a value of 0.15-0.25 cm s$^{-1}$ for 273 K. Therefore, in this work, we assume $SO_2$ dry deposition velocity over snow and ice is 0.01 cm s$^{-1}$ when temperatures are lower than 253 K.

3. **Results and discussions**

To investigate the impacts of updated wet processes on global simulation of aerosols and aerosol precursors, we run GEOS-Chem for 3 cases: (1) standard Geos-Chem version 12.6.0, called GC12; (2) the same as case GC12 except using wet scavenging described in the work of Luo et al. (2019), and this case is named L2019; (3) the same as the case L2019 except considering the updated wet processes described in section 2, and this case is called WETrev. All simulations are run with 2º×2.5º horizontal resolution and 47 layers from surface to 0.01 hPa. Emission over Europe is based on the EMEP inventory. Other emissions are produced by the default setting of HEMCO (Keller et al., 2014) for all simulations presented in this work.

**3.1 Comparison with surface monitoring networks over the US, Europe, and Asia**

To validate model results with surface monitoring networks, we use observational data taken at USEPA, CASTNET, AMoN, IMPROVE and CSN, EMEP and EANET sites. The criterion of observations used for model validation is that valid data are available for every month in 2011. For EANET observations, due to too much missing data, the criterion is loosened to monthly mean data available for each month during a 3-year period (2010-2012). Seto et al. (2007) pointed out that EANET observations at urban sites are much higher than those at remote sites. Since the number of the Asian sites is very limited, to make the validation more appropriate,

only remote and rural sites are used for model validation. Table 2 shows number of sites with observations and number of sites satisfying these criteria. Figure 1 and Table 3 present the comparisons of observed secondary inorganic aerosol precursors and secondary inorganic aerosols at surface monitoring networks and the simulated mass concentrations by the GC12, L2019, and WETrev cases described above. As shown in Fig. 1 (a-c), simulated $SO_2$ for the 3 cases is lower than observed values over the US but higher than the observations over Europe and Asia. Over the US, simulated $SO_2$ is ~ 20% lower than observations. One possible reason is that a large amount of USEPA observations are located at urban regions where $SO_2$ concentrations are much higher than rural and remote regions. There were 288 USEPA sites with valid data in each month of 2011. Only 69 of these sites were with the mark of 'Not in a city'. After considering the updates of wet scavenging by L2019, NMBs are increase from 20% to 23% over the US, reduced from 74% to 59% over Europe, and reduced from 63% to 43% over Asia, respectively. Considering of updated wet processes in this work further reduces NMBs to 51% in Europe and 23% in Asia, respectively.

Figure 1 (d-f) are the results for nitric acid. NMBs of simulated nitric acid by GC12 for the US, Europe, and Asia are 78%, 107%, and 121%, respectively. G12 significantly overestimates surface mass concentration of nitric acid at these regions. Simulations by L2019 and WETrev indicate that wet scavenging is the dominant process causing the overestimation of nitric acid in GEOS-Chem. NMBs of simulated nitric acid in WETrev for the US, Europe, and Asia are reduced to 0.9%, -0.7%, and -21%, respectively. We also notice that WETrev underestimates nitric acid at low temperatures for the US and Europe sites. These underestimates may be associated with the updated uptake coefficients by Holmes et al. (2019) for heterogeneous chemistry. If we switch back to the old heterogeneous chemistry in GEOS-Chem version 12.5, the underestimation of nitric acid at low temperatures is reduced (not shown). Figure 1 (g-i) show that the biases of model simulated ammonia by the 3 cases over the 3 regions are small. Since the increasing ammonia wet deposition is compensated by less equilibrium partitioning with decreased nitric acid in the air, wet processes show relatively small impact on the simulation of ammonia.

Figure 1 (j-l) are observed and simulated sulfate over the US, Europe, and Asia. NMBs of the GC12 case over the 3 regions are -1.1%, 6.9%, and 5.5%, respectively. The application of updates to wet scavenging in L2019 leads to a significant underestimation of sulfate during

winter time, reaching up to 50% over the 3 regions. Based on our investigation, we found that the missing of aqueous phase chemistry in mixed cloud appears to be the main reason of underestimated sulfate at low temperatures. As we discussed in section 2, aqueous phase chemistry in GC12 is only simulated when temperatures are higher than 258 K. Conversely, in WETrev case, the temperature limitation of aqueous phase chemistry is extended from 258 K to 237 K. This change allows aqueous phase chemistry to be simulated when temperatures are low. After employing the new approaches of cloud water pH and aqueous phase cloud fraction calculation, NMBs of the WETrev case at the 3 regions are -10%, 4.3%, and 6.3%, respectively. It significantly reduces the bias shown in the L2019 case. The absence of aqueous phase hydroxymethanesulfonate chemistry may also be a potential reason for the remaining model biases with sulfate, but this is not explored here (Moch et al., 2018). NMB of sulfate simulated by WETrev in the US is higher than that of GC12. However, the good agreement between GC12 sulfate and the observation can be attributed to the coincidental offsets of the higher sulfate mass due to the underestimation of sulfate wet scavenging and the lower sulfate mass due to the absence of aqueous phase chemistry in mixed cloud and hydroxymethanesulfonate chemistry. As shown in Figure 1 (m-r), simulated nitrate and ammonium in the GC12 case over the 3 regions are much higher than observations. As discussed in Luo et al. (2019), the overestimation is associated with the underestimation of rainout and washout of nitric acid and nitrate. Updated wet scavenging in L2019 successfully reduces NMBs of nitrate over the 3 regions from 126% to 10%, 127% to 7.5%, and 269% to 47%, respectively. NMBs of ammonium over the 3 regions are reduced from 45% to -13%, 90% to -7.3%, and 167% to 42%, respectively. Updated wet processes in this work show relatively small impact on simulated nitrate and ammonium surface mass concentrations over the 3 regions.

For simplicity, the WETrev case includes all updates to wet processes as described in Section 2. To understand the contribution of various updates to the overall changes in the predicted concentrations of aerosols and aerosol precursors, we carry out five numerical sensitivity study cases (RO, WO, RP, DD, and AC). RO case is the same as case WETrev except using rainout rate in GC12; WO case is the same as case WETrev except using washout rate in GC12; RP case is the same as case WETrev except assuming pH of rainwater for wet scavenging is 4.5; DD case is the same as case WETrev except using dry deposition treatment in GC12; and

AC case is the same as case WETrev except using aqueous phase chemistry treatment in GC12.

Relative contributions to the changes are calculated as:

$$RC_i = \frac{\sum\limits_{j=1}^{nsite}\left|C_{i,j}-C_{WETrev,j}\right|}{\sum\limits_{j=1}^{nsite}\left|C_{RO,j}-C_{WETrev,j}\right|+\sum\limits_{j=1}^{nsite}\left|C_{WO,j}-C_{WETrev,j}\right|+\sum\limits_{j=1}^{nsite}\left|C_{PR,j}-C_{WETrev,j}\right|+\sum\limits_{j=1}^{nsite}\left|C_{DD,j}-C_{WETrev,j}\right|+\sum\limits_{j=1}^{nsite}\left|C_{AC,j}-C_{WETrev,j}\right|} \quad, (16)$$

where RC is the relative contribution (%), C is simulated surface mass concentration ($\mu g\ m^{-3}$), i

is the numerical sensitive study case index (e.g. when i=1, $C_{i,j}$ refers to $C_{RO,j}$), j is the site index.

Relative contributions of RO, WO, RP, DD, and AC to the changes of January and July

surface concentrations over the USA, Europe, and Asia sites are summarized in Table 4. In the

US, the changes of $SO_2$ are mainly caused by DD and AC whose contributions are up to 54.2%

and 25.0% in January and 50.5% and 22.3% in July. Rainout and washout both show a relatively

small impact on the changes of $SO_2$. In contrast, rainout and washout are important to the

changes of nitric acid, sulfate, nitrate, and ammonium. The contribution of wet scavenging to the

changes of these species exceeds 50% in both January and July. For nitric acid, nitrate, and

ammonium, the contribution of wet scavenging can be as high as 70-90%. For sulfate, AC also

plays an important role with relative contributions in January and July of 29.5% and 17.5%,

which is comparable to the contributions of RO and WO. For ammonia, most of the changes are

caused by DD and AC, with the sum of the 2 processes contributing > 50% of the changes. The

contribution of RP to $SO_2$, sulfate, ammonia, and ammonium is small in January and large in

July. In July the contribution of RP to $SO_2$, sulfate, ammonia, and ammonium is 8.5%, 4.4%,

13.4%, and 4.1%, respectively. The relative contribution from RO, WO, RP, DD, and AC at the

sites over Europe and Asia are similar to those over the US (Table 4).

Figure 2 is a comparison of observed BC and OC over the US and Europe. Simulated BC

over the US is close to observations except for a 10-20% underestimate during summer and fall.

The underestimate is likely associated with the underestimated wildfire emissions in the western

US (Mao et al., 2015). Simulated OC over the US is close to observations during summer but 50-

60% lower than observations during spring and fall. GEOS-Chem (all three cases) significantly

underestimates BC and OC over Europe and the possible reasons behind the bias remain to be

investigated. NMBs of the BC and OC in Europe are up to -37% and -61%, respectively. The

differences of simulated BC and OC in the 3 cases are small for the US and Europe which

indicates wet processes have a small impact on the simulation of BC and OC in these regions.

The small impact of wet processes on BC in the US and Europe is because 80% of emitted BC is assumed to be hydrophobic aerosol which needs 1.15 days to be converted to hydrophilic BC. Updated wet processes have little impact on hydrophobic aerosol in the lower troposphere where wet scavenging is dominated by warm clouds. OC consists of primary organic aerosol (POA) and SOA which is formed through the oxidation of organic gaseous precursors. Due to low dissolution of POA and organic gaseous precursors in water, wet processes also have little impact on these species.

Wet deposition of simulated $SO_2$+SO4, $HNO_3$+NIT, and $NH_3$+NH4 are compared with NTN observations over the US (Fig. 3), EMEP observations over Europe (Fig. 4), and EANET observations over remote region in Asia (Fig. 5). The criteria of observations used for model validation are (1) valid data are available for each month in 2011and (2) the difference between observed and simulated monthly precipitation is within a factor of 4 (Paulot et al., 2014). Number of sites with observations and number of sites satisfying these criteria are shown in Table 5. For the comparison shown in Table 6, model simulated wet depositions are corrected following Paulot et al. (2014) to remove bias due to precipitation. As shown in Figure 3 and Table 6, GC12 underestimates $SO_2$+SO4 wet deposition over the US and Europe. NMBs of $SO_2$+SO4 wet deposition simulated by GC12 over the two region are -21% and -46%, respectively. After considering the updated wet processes in WETrev, NMBs of $SO_2$+SO4 wet deposition are reduced to -9.0% over the US and -6.2% over Europe, respectively. However, all the three cases significantly underestimate $SO_2$+SO4 wet deposition over Asia. One possible reason is that GEOS-Chem may underestimate eruptive volcanic emission nearby the four Japanese sites. For $HNO_3$+NIT wet deposition over the US, the values simulated by GC12 are close to observations, while the values simulated by WETrev are ~ 2 times higher than observations. However, wet deposition data are collected weekly at NTN sites. It is hard to estimate the uncertainty due to the evaporation of HNO3 from the collected precipitation water. Over Europe and Asia, wet deposition fluxes are observed daily at most of EMEP and EANET sites. The values of $HNO_3$+NIT wet deposition simulated by GC12 are lower than observations, while the values simulated by WETrev are higher than observations. For $NH_3$+NH4, GC12 underestimates wet deposition over the US, Europe, and Asia. NMBs over the 3 regions are -10%, -33%, and -10%, respectively. NMBs of $NH_3$+NH4 wet deposition simulated by WETrev are reduced to -7.7% over Europe and -2.5% over Asia, respectively.

## 3.2 Comparison of SO$_2$, sulfate and BC mass concentrations at Arctic sites

We also study the impact of updated wet processes on SO$_2$, sulfate and BC surface mass concentrations at several Arctic sites where measurements are available. Figure 6 shows the comparison of SO$_2$ at Nord (81.6ºN, 16.7ºW) and Zeppelin (78.9ºN, 11.9ºE). GC12 matches well with the observed SO$_2$ at Nord but overestimates SO$_2$ at Zeppelin in January and December by a factor of 3. The updated wet scavenging (yellow line) shows a small impact on simulated SO$_2$ in the Arctic., with simulated SO$_2$ reduced slightly during winter and spring. In WETrev, we assumed SO$_2$ dry deposition velocity is 0.01 cm s$^{-1}$ when temperatures are lower than 253 K. WETrev slightly enhances SO$_2$ at the higher latitude site Nord during winter. At Zeppelin, temperature in December is higher than that in January and February, and SO$_2$ concentration is enhanced due to the modification of dry deposition in this work. However, there is more aqueous phase chemistry in December which consumes the enhanced SO$_2$. By switching from GC12 to WETrev, NMB of SO$_2$ is increased from -23% to 32% at Nord and decreased from 27% to 22% at Zeppelin. Figure 7 compares the observed and simulated sulfate and BC at Alert (82.5ºN, 62.5ºW), Barrow (71.3ºN, 156.6ºW), and Zeppelin. Observations at the three sites show that both sulfate and BC are high in spring and low in summer. The model simulations generally capture seasonal variation at these Arctic sites. However, GC12 overestimates sulfate mass concentration at the 3 sites by a factor of 2-3. Simulated BC by GC12 is 50% lower than observation at Alert during winter and spring and a factor of 2 higher than observations at Barrow and Zeppelin during winter. Updated wet scavenging significantly impacts simulated sulfate and BC in Arctic regions. Simulated sulfate by L2019 is much closer to observations except for a 50% underestimation at Alert during winter and spring, while simulated BCs at the 3 Arctic sites by L2019 is much lower than observations. The comparison with model results from WETrev shows the underestimation of sulfate at Alert during spring is compensated by considering aqueous phase chemistry in mixed clouds. Most of BC at Arctic regions is transported from middle-low latitude source regions with open fire and anthropogenic emissions (Xu et al., 2017), and during the long-range transport hydrophobic BC is aged and converted to hydrophilic BC. The assumption of reduced hydrophilic BC rainout efficiency in the WETrev case increases simulated BC mass concentration and enhances agreement with observations at these Arctic sites.

NMBs of BC are reduced from -67% to -40% at Barrow and from -75% to -46% at Zeppelin due to the switch from L2019 to WETrev.

### 3.3 Vertical profiles of nitric acid and aerosols: Comparison with ATom-1 and ATom-2 aircraft measurements

To evaluate the impact of updated wet processes on simulated vertical profiles of aerosols and aerosol precursors, we compare simulated nitric acid and aerosols for the 3 cases with the aircraft measurements of ATom-1 in July-August 2016 and ATom-2 in January-February 2017 (Jimenez et al., 2019; Wofsy et al., 2018) over the Northern Hemisphere (Fig. 8) and the Southern Hemisphere (Fig. 9). Nitric acid was measured by Chemical Ionization Mass Spectrometer, while aerosols were measured by CU Aircraft High-Resolution Time-of-Flight Aerosol Mass Spectrometer (Hodzic et al., 2020). The work of Brock et al. (2019) indicated that there is very good quantitative agreement between AMS and volume data. For ATom data, OC is calculated by OA_PM1_AMS/OAtoOC_PM1_AMS. For model, we used 1.8 for SOAs. Flight tracks over the land or in the stratosphere are filtered out for the comparison (see Figure S1 in supporting materials for flight tracks of ATom-1 amd ATom-2). We filter out the flight tracks over the land is because ATom observations over the land, whose values vary greatly, only account for 28% of total measurements. The exclusion of these data makes the comparison more appropriate. Vertical profiles of nitric acid and aerosols over the land, which are similar to Fig. 8 and 9, are shown in Figure S2.

As shown in Figure 8, GC12 overestimates nitric acid and underestimates black carbon and organic carbon over the Northern Hemisphere during both ATom-1 and ATom-2. NMBs of the 3 species are 66%, -77%, and -55% during ATom-1 and 163%, -10%, and -27% during ATom-2. GC12 simulated sulfate and ammonium match well with observations during ATom-1 but are much higher than observations during ATom-2 whose values are high up to 78% for sulfate and 217% for ammonium. After considering the updated wet scavenging in L2019, the overestimates of nitric acid, sulfate, and ammonium during ATom-2 and nitric acid during ATom-1 are reduced to 5%, -11%, -30%, and -36%, respectively. However, L2019 significantly underestimates nitric acid at the upper troposphere where pressure is lower than 300 hPa. As we mentioned earlier, L2019 may overestimate cold cloud wet scavenging of nitric acid due to treatment of cold cloud rainout of nitric acid as same as water-soluble aerosol with 100% rainout

efficiency. With updated cold cloud scavenging in WETrev, the bias of nitric acid simulated by L2019 at the upper troposphere is reduced during ATom-2 and is enhanced during ATom-1. This indicates further understanding regarding ice uptake and removal of nitric acid are needed. Nitric acid concentrations simulated by WETrev between 500 hPa and 300 hPa are much lower than those simulated by L2019 and GC12. This is because WETrev considers washouts of nitric acid by snow and ice which were absent in L2019 and GC12. Figure 8 (g) shows the impact of updated aqueous phase chemistry in mixed clouds on the sulfate vertical profile. Considering aqueous phase chemistry in mixed clouds significantly enhances sulfate mass concentration within the range of 700-500 hPa during ATom-2 which makes the simulated sulfate much closer to observed values. Figures 8 (d) and (i) indicate that the impact of updated wet scavenging on the black carbon vertical profile during ATom-2 is more obvious than that during ATom-1. This is because there is much less black carbon emitted from open fires in January than there is in July. Black carbon observed during ATom-2 is dominated by hydrophilic black carbon which is more affected by wet scavenging processes, while black carbon observed during ATom-1 is dominated by hydrophobic black carbon. Updated wet scavenging shows a small impact on organic carbon vertical profiles during both ATom-1 and ATom-2. Figure 9 shows comparisons over the Southern Hemisphere. Updated wet scavenging reduces overestimated nitric acid especially during ATom-1 period. NMB is reduced from 80% to -25%. For sulfate, ammonium, black carbon, and organic carbon, the differences among the 3 cases are relative small. NMBs of WETrev for these species are larger than those of GC12. All cases significantly underestimate black carbon from open fire and organic carbon in the upper troposphere. Based on the comparisons with ATom-1 and ATom-2 measurements, it is clear that the updated wet process treatments in this work and L2019 can improve the agreements of simulated and observed vertical profiles of nitric acid (Fig. 8a, Fig. 8f, Fig. 9a, and Fig. 9f). The simulated of winter time sulfate and ammonium in the Northern Hemisphere are also improved by WETrev.

### 3.4 Impact on global distributions of surface mass concentrations

The impacts of updated wet process treatments on global simulation of surface mass concentrations are shown in Figures 10-14. Figures 10-12 show simulated surface mass concentrations of secondary inorganic aerosol precursors ($SO_2$, nitric acid, and ammonia), secondary inorganic aerosols (sulfate, nitrate, and ammonium), primary inorganic aerosols (sea-

salt, dust, and black carbon), and organic carbon (primary organic aerosol and secondary organic aerosol) simulated by GC12 case and WETrev case, while figures 13-14 are the percentage differences.

As shown in Figure 10, high values of secondary inorganic aerosol precursors are mainly located over continental regions with high anthropogenic and natural emissions. After considering the updated wet process treatments in this study, global mean surface mass concentrations (GMSMC) of $SO_2$, nitric acid, and ammonia are changed from 0.73 µg m$^{-3}$, 0.56 µg m$^{-3}$, 0.32 µg m$^{-3}$ to 0.75 µg m$^{-3}$, 0.26 µg m$^{-3}$, 0.42 µg m$^{-3}$, respectively. The updated wet process treatments slightly impact GMSMC of $SO_2$ but strongly impact GMSMC of nitric acid. The impact on ammonia is small over land but large over ocean. The weak impact of the updated wet process treatments on $SO_2$ is because its wet removal is dominated by aqueous phase chemistry. The strong impact of the updated wet process treatments on ammonia over ocean is due to the changes of rainwater pHs over remote regions whose values are higher than the assumed 4.5 rainwater pH in GC12. Some large changes of surface mass concentration at Arctic and Antarctic regions, as shown in Figure 13 (a-c), are associated with the updated treatments of wet surface uptake during dry deposition at snow and ice. However, due to low mass concentrations for Arctic and Antarctic regions, their impacts on GMSMC are small. The updated wet process treatments significantly impact GMSMC of secondary inorganic aerosols whose water solubility is high. After considering the updated wet process treatments, GMSMC of sulfate, nitrate, and ammonium are changed from 0.84 µg m$^{-3}$, 0.42 µg m$^{-3}$, 0.33 µg m$^{-3}$ to 0.74 µg m$^{-3}$, 0.21 µg m$^{-3}$, 0.26 µg m$^{-3}$, respectively. Their global mean relative changes are high up to -25%, -53%, and -22%, respectively. Most of the reductions of these species happen at middle-high latitude regions with high mass concentrations.

Figures 12 and 14 show the impact of updated wet process treatments on primary inorganic aerosols and organic carbon. It is clear that the updated wet process treatments have little impact on GMSMCs of these species. For sea salt, its high concentrations are mainly located at middle latitude regions in both the Northern Hemisphere and Southern Hemisphere where in cloud condensation water values are close to the assumed constant value in GC12. Therefore, the differences of wet scavenging in GC12 and WETrev cases at these regions are small. For dust, due to its low water solubility, the updated wet processes show a small impact in the lower troposphere where wet scavenging is dominated by warm clouds. Most of black carbon

and organic carbon are emitted as hydrophobic aerosols and then converted to be hydrophilic aerosols due to aging. Therefore, the updated wet process treatments show only a small impact at source regions but show a strong impact for remote regions.

**4. Summary**

In this study, we updated aqueous phase chemistry and wet scavenging for $SO_2$ and sulfate, rainout efficiencies for warm, mixed, and cold clouds, empirical washout by rain and snow, and wet surface uptake during dry deposition in GEOS-Chem version 12.6.0. Systematic validations of simulated aerosols and aerosol precursors with ground based monitoring networks over the US, Europe, and Asia, in-site observations at Arctic for surface mass concentrations and aircraft measurements during ATom-1 and ATom2 for their vertical profiles were presented. Based on these validations, we found:

(1) The model results with the updated treatment of wet processes agree better with measurements for most species in different regions, especially for nitric acid, nitrate, and ammonium whose NMBs were improved, respectively, from 78%, 126%, and 45% to 0.9%, 15%, and 4.1% over US sites, from 107%, 127%, and 90% to -0.7%, 4.2%, and 16% over Europe sites, and from 121%, 269%, and 167% to -21%, 37%, and 86% over Asia remote region sites;

(2) Comparing to Luo et al. (2019), the updated aqueous phase chemistry and wet scavenging of $SO_2$ and sulfate significantly improve the agreement of simulated $SO_2$ and sulfate over the US, Europe and Asia remote region, especially during the winter time. NMBs of sulfate in the 3 regions are reduced from -30%, -33%, and -36% to -10%, 4.3%, and 6.3%;

(3) The updated wet process treatments significantly improve the performance of sulfate wet deposition simulation over the US and Europe. NMBs are reduced from -35% to -9% over the US and from -46% to -6.2% over Europe, respectively;

(4) The updated rainout efficiencies enhance BC mass concentration for remote regions and successfully reduce the bias between simulation and observation at Arctic sites. NMBs of BC are reduced from -67% to -40% at Barrow and from -75% to -46% at Zeppelin due to the switch from L2019 to WETrev;

(5) Cold cloud scavenging plays important roles in the simulation at the upper troposphere, especially for nitric acid.

(6) The updated wet surface uptake during dry deposition changes the performance of simulated $SO_2$ at Arctic sites. NMB of $SO_2$ is increased from -23% to 32% at Nord and decreased from 27% to 22% at Zeppelin.

Wet processes are important for atmospheric chemistry modeling. Our study indicates that the updated wet process treatments introduced in this study have strong impacts on global means of water soluble aerosols and aerosol precursors such as nitric acid, sulfate, nitrate, and ammonium. The updated wet process treatments exhibit relatively small impacts on the simulated global means of $SO_2$, dust, sea salt, black carbon, and organic carbon. Although there are clear improvements derived from the updated treatment of wet processes, there still exist limitations of the work presented in this study. For example, washout efficiencies of water soluble species such as $SO_2$ and ammonia are sensitive to rain water pH values. In this study, we simply assumed rainwater pHs for rainout and washout are cloud pH at where rainout occurs and rainwater-mass-weighted cloud pH above where washout occurs, respectively. However, rain water pH needs to be calculated by tracing the cloud process and precipitation process of rain water lifecycle. The impact of traced rain water pH on wet scavenging needs to be further investigated.

Code and data availability. The code of GEOS-Chem 12.6.0 is available through the GEOS-Chem distribution web-page http://wiki.seas.harvard.edu/geos-chem/index.php/GEOS-Chem_12. The updated wet process code can be obtained by contacting the author directly. All measurement data are publicly available. USEPA data are download from https://www.epa.gov/outdoor-air-quality-data; CASTNET, AMon, IMPROVE, CSN data are download from http://views.cira.colostate.edu/fed/; NTN data are download from http://nadp.slh.wisc.edu/data/ntn/ntnAllsites.aspx; EMEP date are download from http://ebas.nilu.no/default.aspx and https://projects.nilu.no//ccc/emepdata.html; EANET data are download from https://monitoring.eanet.asia/document/signin; ATom data are download from https://espoarchive.nasa.gov/archive/browse/atom.

Author contributions. GL and FY proposed and implemented the improved wet processes schemes and validated model simulations with surface observations and ATom aircraft measurements. JM provided the new cloud pH approach in GEOS-Chem. All authors contributed to the writing and editing of the paper.

Competing interests. The authors declare that they have no conflict of interest.

Acknowledgments. This work is supported by NYSERDA under contract 137487, NASA under grant NNX17AG35G, and NSF under grant 1550816. The authors thank Daniel J. Jacob, Harvard University, whose comments and suggestions greatly helped improve and clarify this paper. We would like to acknowledge the United States Environmental Protection Agency (USEPA), the Interagency Monitoring of Protected Visual Environments (IMPROVE), the Chemical Speciation Network (CSN), the Clean Air Status and Trends Network (CASTNET), the Ammonia Monitoring Network (AMoN), National Trends Network (NTN), the European Monitoring and Evaluation Programme (EMEP), and the Acid Deposition Monitoring Network in East Asia (EANET) for the in-site measurement data. We would like to acknowledge the Atmospheric Tomography Mission (ATom) for the aircraft measurement data (https://daac.ornl.gov/ATOM/campaign/). GEOS-Chem is a community model maintained by the GEOS-Chem Support Team at Harvard University.

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

1    Table 1. List of Λ and b values in equation 15 for rain and snow washout parameterizations.

| | Rain | | Snow | |
|---|---|---|---|---|
| | T>268 K | | 248 K<T<268 K | |
| | Λ | $b$ | Λ | $b$ |
| GC12 | | | | |
| $HNO_3$ | $2.8\times10^{-5}$ | 1.0 | 0 | 0 |
| Coarse aerosol | $2.6\times10^{-4}$ | 0.79 | $4.2\times10^{-4}$ | 0.96 |
| Fine aerosol | $4.3\times10^{-6}$ | 0.61 | $8.8\times10^{-6}$ | 0.96 |
| This work | | | | |
| $HNO_3$ | $3\times10^{-3}$ [‡] | 0.62 [‡] | $3\times10^{-3}$ [‡] | 0.62 [‡] |
| Coarse aerosol | $2\times10^{-4}$ [†] | 0.85 [†] | $2\times10^{-3}$ [†] | 0.7 [†] |
| Hydrophobic fine aerosol | $5\times10^{-7}$ [†] | 0.7 [†] | $1\times10^{-5}$ [†] | 0.66 [†] |
| Hydrophilic fine aerosol | $1\times10^{-5}$ [*] | 0.7 [†] | $2\times10^{-4}$ [*] | 0.66 [†] |

2    [†] from Wang et al. (2014) assuming fine aerosol with diameter of 100 nm and coarse aerosol

3    with diameter of 6 μm; [‡] from Luo et al. (2019); [*] this work.

1    Table 2. Number of sites with surface concentration observation (NVO) and number of sites

2    satisfying criterion (NSC) at surface monitoring networks in the US, Europe, and Asia.

|  | USA | | Europe | | Asia | |
|---|---|---|---|---|---|---|
|  | NVO | NSC | NVO | NSC | NVO | NSC |
| SO2 | USEPA | | EMEP | | EANET | |
|  | 464 | 288 | 42 | 20 | 14 | 3 |
| HNO3 | CASTNET | | EMEP | | EANET | |
|  | 84 | 77 | 25 | 8 | 25 | 5 |
| NH3 | AMoN | | EMEP | | EANET | |
|  | 53 | 17 | 40 | 15 | 25 | 10 |
| SO4 | IMPROVE+CSN | | EMEP | | EANET | |
|  | 371 | 214 | 52 | 21 | 25 | 9 |
| NIT | IMPROVE+CSN | | EMEP | | EANET | |
|  | 371 | 213 | 66 | 22 | 25 | 8 |
| NH4 | IMPROVE+CSN | | EMEP | | EANET | |
|  | 371 | 178 | 66 | 24 | 25 | 9 |
| BC | IMPROVE | | EMEP | | | |
|  | 168 | 122 | 11 | 5 | | |
| OC | IMPROVE | | EMEP | | | |
|  | 168 | 118 | 11 | 5 | | |

Table 3. Observed and simulated annual mean surface concentrations of aerosols and aerosol precursors in the US, Europe, and Asia. Comparisons include annual mean surface concentrations (M, µg m⁻³), normalized mean bias (NMB, %), and correlation coefficient ($r$, when # of samples > 10) between observed and simulated annual mean values for the 8 species by G12, L2019, and WETrev cases.

| | | USA | | | | Europe | | | | Asia | | |
|---|---|---|---|---|---|---|---|---|---|---|---|---|
| | | G12 | L2019 | WETrev | | G12 | L2019 | WETrev | | G12 | L2019 | WETrev |
| SO2 | M: 5.61 | 4.48 | 4.29 | 4.32 | M: 1.36 | 2.36 | 2.16 | 2.05 | M: 2.51 | 4.08 | 3.58 | 3.08 |
| | NMB | -20 | -23 | -23 | NMB | 74 | 59 | 51 | NMB | 63 | 43 | 23 |
| | $r$ | 0.49 | 0.49 | 0.48 | $r$ | 0.53 | 0.50 | 0.50 | $r$ | | | |
| HNO3 | M: 0.83 | 1.47 | 0.83 | 0.83 | M: 0.67 | 1.40 | 0.66 | 0.67 | M: 0.86 | 1.90 | 0.64 | 0.68 |
| | NMB | 78 | 0.9 | 0.9 | NMB | 107 | -2.4 | -0.7 | NMB | 121 | -26 | -21 |
| | $r$ | 0.57 | 0.59 | 0.60 | $r$ | | | | $r$ | | | |
| NH3 | M: 1.00 | 1.02 | 1.04 | 1.21 | M: 0.83 | 0.84 | 0.91 | 1.07 | M: 0.96 | 0.95 | 0.88 | 1.06 |
| | NMB | 2.6 | 4.4 | 21 | NMB | 0.9 | 8.7 | 28 | NMB | -1.7 | -8.6 | 10 |
| | $r$ | 0.26 | 0.28 | 0.29 | $r$ | 0.90 | 0.89 | 0.91 | $r$ | | | |
| SO4 | M: 1.30 | 1.29 | 0.91 | 1.17 | M: 1.29 | 1.38 | 0.87 | 1.24 | M: 2.63 | 2.77 | 1.69 | 2.46 |
| | NMB | -1.1 | -30 | -10 | NMB | 6.9 | -33 | -4.3 | NMB | 5.5 | -36 | -6.3 |
| | $r$ | 0.92 | 0.92 | 0.92 | $r$ | 0.92 | 0.90 | 0.92 | $r$ | | | |
| NIT | M: 0.71 | 1.60 | 0.78 | 0.81 | M: 1.66 | 3.77 | 1.54 | 1.73 | M: 0.60 | 2.23 | 0.89 | 0.83 |
| | NMB | 126 | 10 | 15 | NMB | 127 | -7.5 | 4.2 | NMB | 269 | 47 | 37 |
| | $r$ | 0.53 | 0.58 | 0.61 | $r$ | 0.85 | 0.86 | 0.86 | $r$ | | | |
| NH4 | M: 0.61 | 0.89 | 0.54 | 0.64 | M: 0.88 | 1.67 | 0.82 | 1.02 | M: 0.58 | 1.55 | 0.82 | 1.08 |
| | NMB | 45 | -13 | 4.1 | NMB | 90 | -7.3 | 16 | NMB | 167 | 42 | 86 |
| | $r$ | 0.76 | 0.79 | 0.79 | $r$ | 0.79 | 0.81 | 0.81 | $r$ | | | |
| BC | M: 0.20 | 0.18 | 0.16 | 0.17 | M: 0.51 | 0.38 | 0.32 | 0.34 | | | | |
| | NMB | -7.0 | -20 | -14 | NMB | -25 | -37 | -32 | | | | |
| | $r$ | 0.54 | 0.54 | 0.54 | $r$ | | | | | | | |
| OC | M: 1.01 | 0.80 | 0.68 | 0.72 | M: 1.97 | 1.00 | 0.77 | 0.85 | | | | |
| | NMB | -20 | -33 | -29 | NMB | -49 | -61 | -57 | | | | |
| | $r$ | 0.63 | 0.65 | 0.65 | $r$ | | | | | | | |

Table 4. Relative contribution (%) of modified rainout (RO), washout (WO), rain pH (RP), dry deposition (DD), and aqueous chemistry (AC) to the changes of January and July surface concentrations at the US, Europe, and Asia sites.

| | USA | | | | | Europe | | | | | Asia | | | | |
|---|---|---|---|---|---|---|---|---|---|---|---|---|---|---|---|
| | RO | WO | RP | DD | AC | RO | WO | RP | DD | AC | RO | WO | RP | DD | AC |
| | January | | | | | | | | | | | | | | |
| SO2 | 5.0 | 15.3 | 0.5 | 54.2 | 25.0 | 11.7 | 24.1 | 12.0 | 19.0 | 33.1 | 3.6 | 15.3 | 0.2 | 27.7 | 53.2 |
| HNO3 | 15.5 | 73.4 | 0.5 | 5.3 | 5.2 | 25.2 | 60.1 | 1.3 | 2.4 | 11.0 | 8.7 | 63.1 | 0.1 | 8.4 | 19.6 |
| NH3 | 7.9 | 23.7 | 1.6 | 30.5 | 36.3 | 9.0 | 20.4 | 31.3 | 14.9 | 24.4 | 3.9 | 7.0 | 5.8 | 26.2 | 57.1 |
| SO4 | 46.6 | 17.3 | 0.4 | 6.2 | 29.5 | 74.3 | 8.5 | 0.9 | 2.1 | 14.3 | 29.4 | 17.5 | 0.1 | 5.8 | 47.3 |
| NIT | 37.7 | 46.7 | 0.7 | 5.3 | 9.6 | 56.5 | 34.1 | 1.4 | 1.5 | 6.5 | 17.4 | 43.9 | 0.3 | 10.7 | 27.6 |
| NH4 | 48.7 | 34.3 | 0.7 | 6.0 | 10.3 | 78.3 | 13.2 | 1.0 | 2.2 | 5.2 | 40.6 | 22.9 | 0.3 | 3.1 | 33.0 |
| | July | | | | | | | | | | | | | | |
| SO2 | 5.6 | 13.1 | 8.5 | 50.5 | 22.3 | 3.0 | 31.3 | 1.3 | 31.0 | 33.4 | 13.3 | 15.9 | 15.2 | 23.5 | 32.1 |
| HNO3 | 5.8 | 91.3 | 0.5 | 2.0 | 0.4 | 5.2 | 93.8 | 0.2 | 0.7 | 0.1 | 11.4 | 86.8 | 0.5 | 1.2 | 0.1 |
| NH3 | 6.7 | 21.7 | 13.4 | 49.9 | 8.2 | 5.7 | 53.2 | 11.5 | 26.6 | 3.0 | 4.8 | 17.9 | 28.7 | 45.0 | 3.6 |
| SO4 | 48.7 | 16.5 | 4.4 | 12.9 | 17.5 | 66.0 | 11.7 | 0.7 | 3.0 | 18.7 | 63.9 | 16.2 | 2.6 | 8.4 | 8.8 |
| NIT | 16.1 | 68.7 | 2.7 | 10.9 | 1.6 | 12.3 | 82.6 | 1.5 | 3.2 | 0.4 | 24.4 | 64.7 | 3.3 | 6.9 | 0.7 |
| NH4 | 35.7 | 36.4 | 4.1 | 13.0 | 10.8 | 27.2 | 63.7 | 1.0 | 2.6 | 5.5 | 52.6 | 29.1 | 3.3 | 8.9 | 6.2 |

Table 5. Number of sites with wet deposition observation (NVO) and number of sites satisfying criterion (NSC) at surface monitoring networks in the US, Europe, and Asia.

| | USA | | Europe | | Asia | |
|---|---|---|---|---|---|---|
| | NTN | | EMEP | | EANET | |
| | NVO | NSC | NVO | NSC | NVO | NSC |
| SO2+SO4 | 250 | 86 | 62 | 25 | 53 | 4 |
| HNO3+NIT | 250 | 86 | 67 | 30 | 53 | 4 |
| NH3+NH4 | 250 | 85 | 64 | 29 | 53 | 4 |

Table 6. Observed and simulated annual mean wet depositions of aerosols and aerosol precursors in the US, Europe, and Asia. Comparisons include annual mean wet depositions (M, kg ha$^{-1}$ year$^{-1}$), normalized mean bias (NMB, %), and correlation coefficient ($r$, when # of samples > 10) between observed and simulated annual mean values by G12, L2019, and WETrev cases. Simulated values at sites were corrected following Paulot et al. (2014) to remove bias due to precipitation.

| | | USA | | | | Europe | | | | Asia | | |
|---|---|---|---|---|---|---|---|---|---|---|---|---|
| | | G12 | L2019 | WETrev | | G12 | L2019 | WETrev | | G12 | L2019 | WETrev |
| SO2+SO4 | M: 10.3 | 6.8 | 8.0 | 9.4 | M: 6.3 | 3.4 | 5.0 | 5.9 | M: 28.6 | 10.3 | 11.1 | 13.2 |
| | NMB | -35 | -23 | -9.0 | NMB | -46 | -21 | -6.2 | NMB | -64 | -61 | -54 |
| | $r$ | 0.81 | 0.79 | 0.81 | $r$ | 0.56 | 0.55 | 0.49 | $r$ | | | |
| HNO3+NIT | M: 9.5 | 9.6 | 18.1 | 19.1 | M: 9.9 | 6.8 | 14.3 | 14.0 | M: 14.6 | 13.3 | 15.8 | 15.5 |
| | NMB | 0.6 | 89 | 100 | NMB | -31 | 45 | 42 | NMB | -9.2 | 8.1 | 6.2 |
| | $r$ | 0.9 | 0.85 | 0.88 | $r$ | 0.84 | 0.59 | 0.64 | $r$ | | | |
| NH3+NH4 | M: 3.6 | 3.2 | 4.0 | 4.2 | M: 3.9 | 2.6 | 3.9 | 3.6 | M: 3.9 | 3.5 | 3.3 | 3.8 |
| | NMB | -10 | 12 | 16 | NMB | -33 | -1.6 | -7.7 | NMB | -10 | -14 | -2.5 |
| | $r$ | 0.85 | 0.87 | 0.85 | $r$ | 0.75 | 0.55 | 0.67 | $r$ | | | |

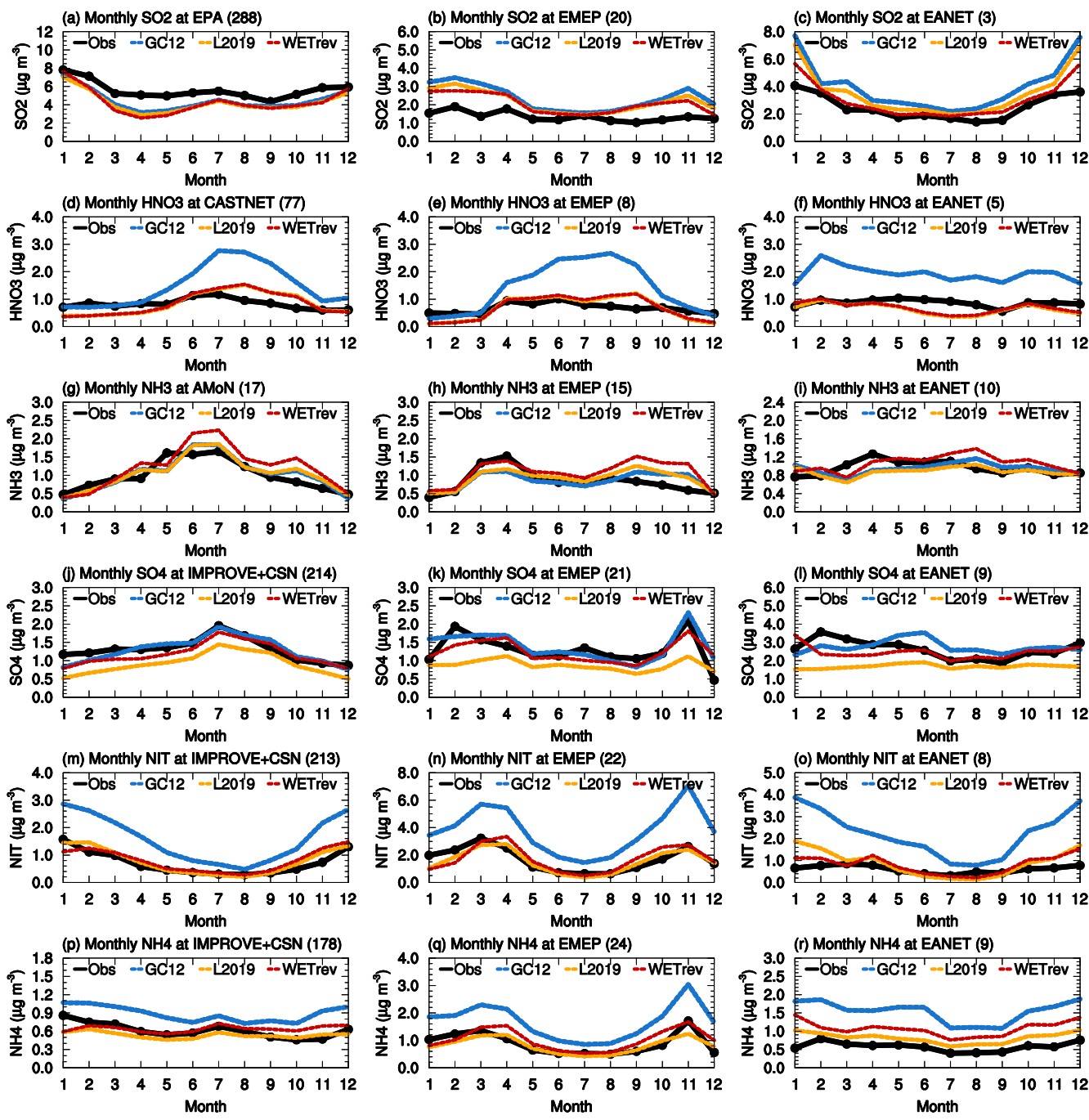

Figure 1. Variations of monthly means for year 2011 showing the comparisons of SO₂, nitric acid, ammonia, sulfate, nitrate, and ammonium surface mass concentrations which are observed over (left column) the US, (center column) Europe, and (right column) Asia sites (black) and simulated by GC12 (blue), L2019 (yellow), and WETrev (red) cases.

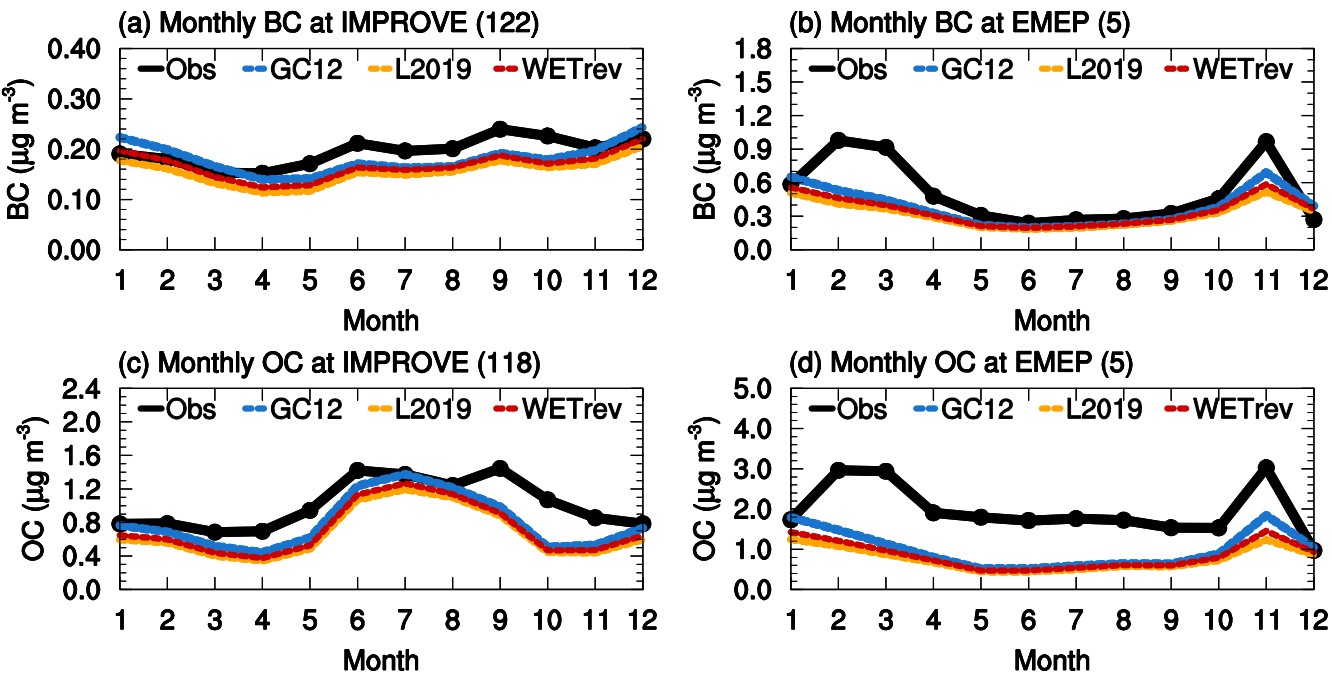

Figure 2. Variations of monthly means for year 2011 showing the comparisons of black carbon and organic carbon surface mass concentrations which are observed over (left column) the US and (right column) Europe sites (black) and simulated by GC12 (blue), L2019 (yellow), and WETrev (red) cases.

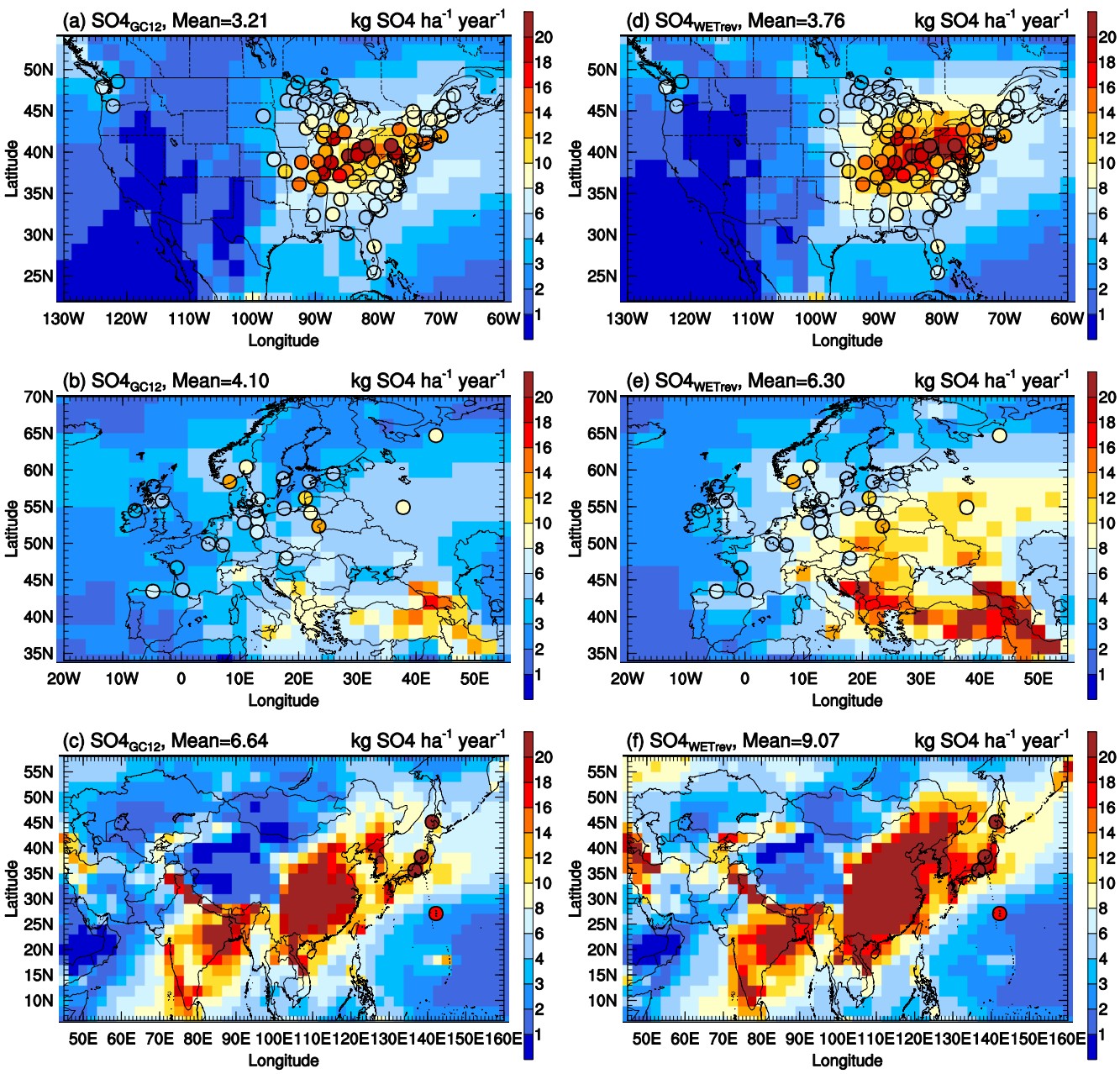

Figure 3. Horizontal distributions of $SO_2+SO_4$ deposition over the US (top), Europe (middle), and Asia (bottom). Filled circles are annual mean wet depositions at NTN, EMEP, and EANET corrected following Paulot et al. (2014) to remove bias due to precipitation.

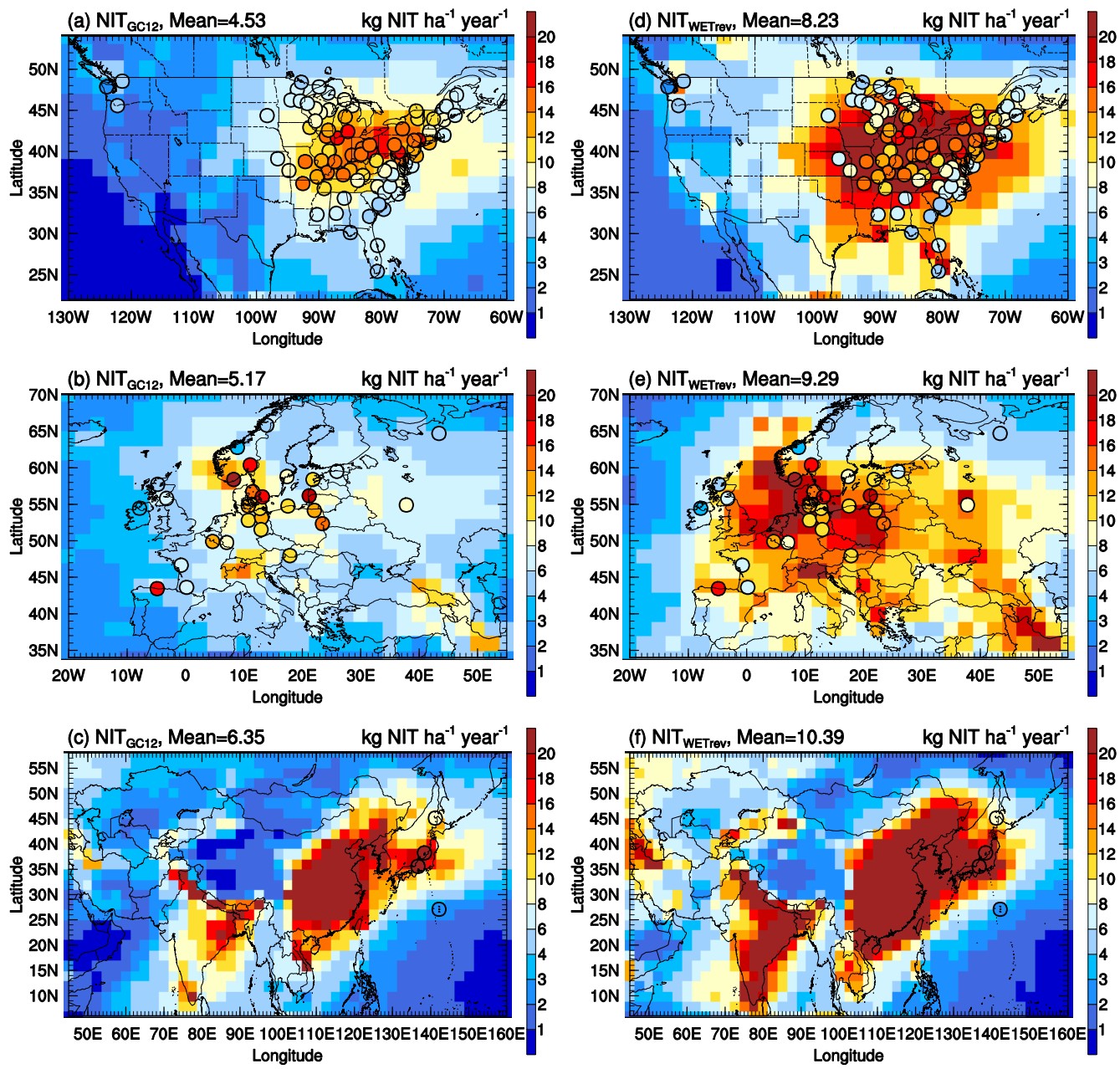

Figure 4. The same as Fig. 3 but for $HNO_3+NIT$.

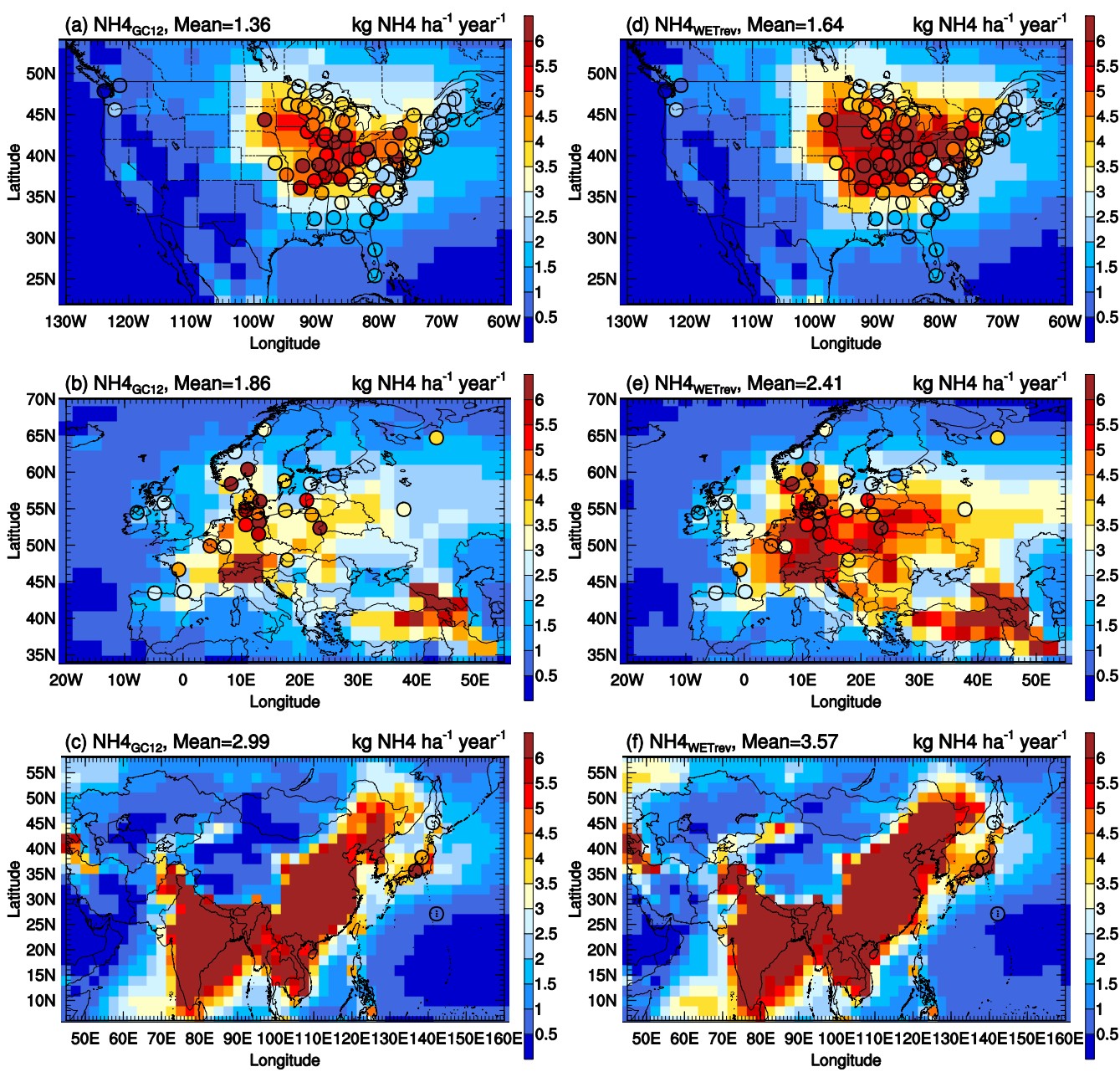

Figure 5. The same as Fig. 3 but for NH3+NH4.

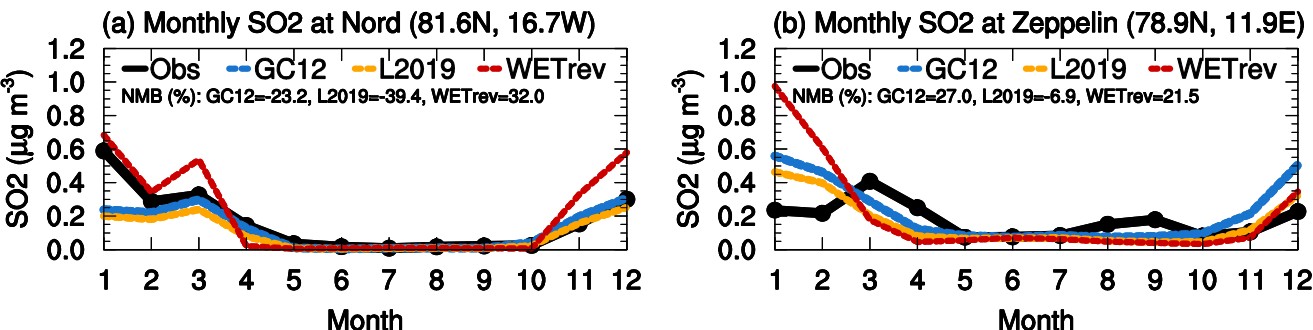

Figure 6. Variations of multiyear monthly means showing the comparisons of SO₂ surface mass concentrations which were observed at (a) Nord (2008-2013) and (b) Zeppelin (2008-2013) sites (black) and simulated (2011) by GC12 (blue), L2019 (yellow), and WETrev (red) cases.

## (a) Monthly SO4 at Alert (82.5N, 62.5W)
NMB (%): GC12=145.0, L2019=-56.3, WETrev=-29.9

## (d) Monthly BC at Alert (82.5N, 62.5W)
NMB (%): GC12=-50.1, L2019=-93.8, WETrev=-81.7

## (b) Monthly SO4 at Barrow (71.3N, 156.6W)
NMB (%): GC12=268.3, L2019=-9.0, WETrev=43.3

## (e) Monthly BC at Barrow (71.3N, 156.6W)
NMB (%): GC12=14.9, L2019=-67.3, WETrev=-39.5

## (c) Monthly SO4 at Zeppelin (78.9N, 11.9E)
NMB (%): GC12=292.7, L2019=3.3, WETrev=31.7

## (f) Monthly BC at Zeppelin (78.9N, 11.9E)
NMB (%): GC12=24.7, L2019=-75.2, WETrev=-46.0

Figure 7. Variations of multiyear monthly means showing the comparisons of (a-c) sulfate and (d-f) black carbon surface mass concentrations which were observed at (top) Alert (2008-2012), (middle) Barrow (2008-2013), and (bottom) Zeppelin (2008-2013) sites (black) and simulated (2011) by GC12 (blue), L2019 (yellow), and WETrev (red) cases.

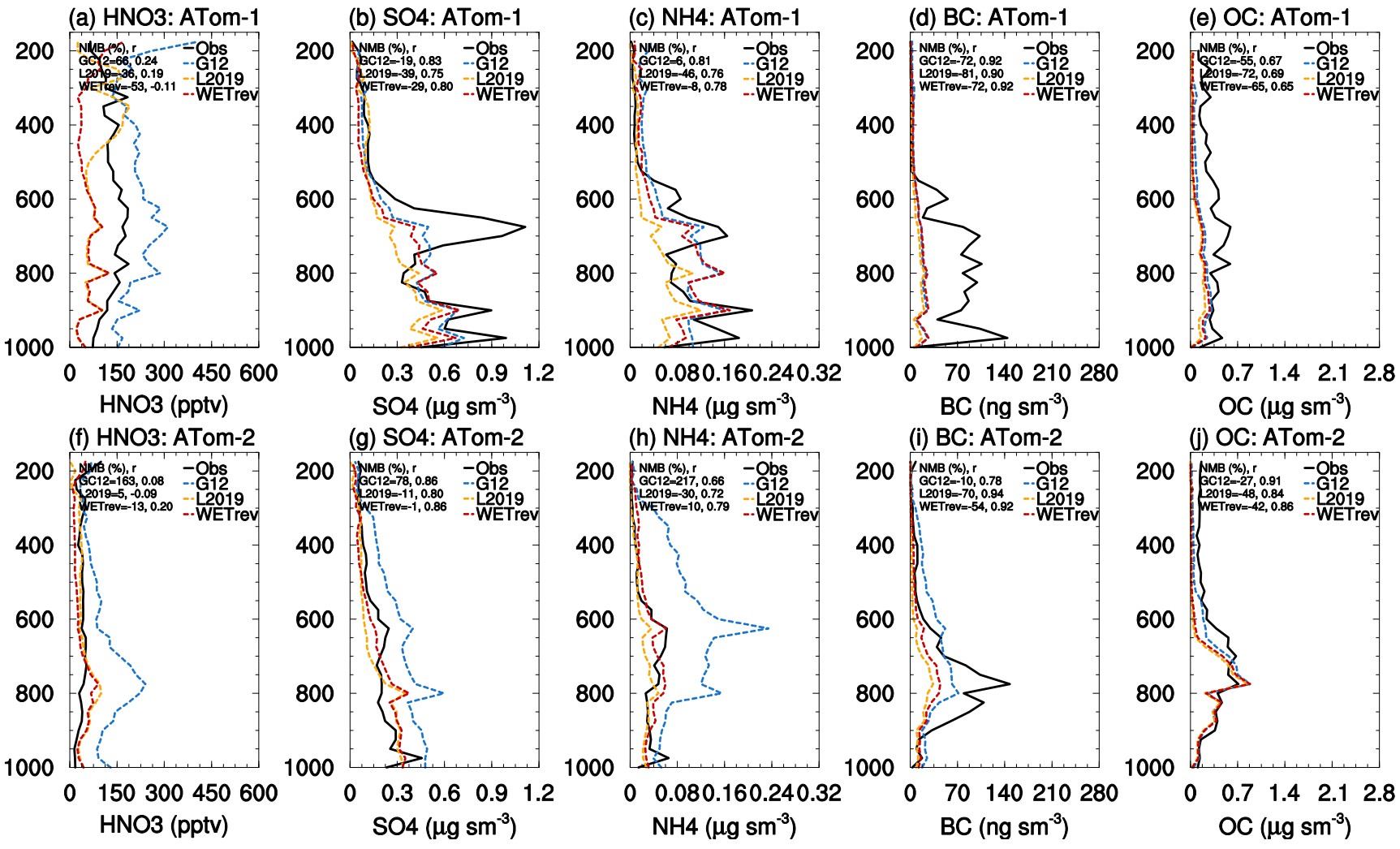

Figure 8. Vertical profiles of nitric acid, sulfate, ammonium, black carbon, and organic carbon from ATom aircraft observations (black, ATom-1: a-e; ATom-2: f-j) and GEOS-Chem simulations by GC12 (blue), L2019 (yellow) and WETrev (red) cases over the Northern hemisphere.

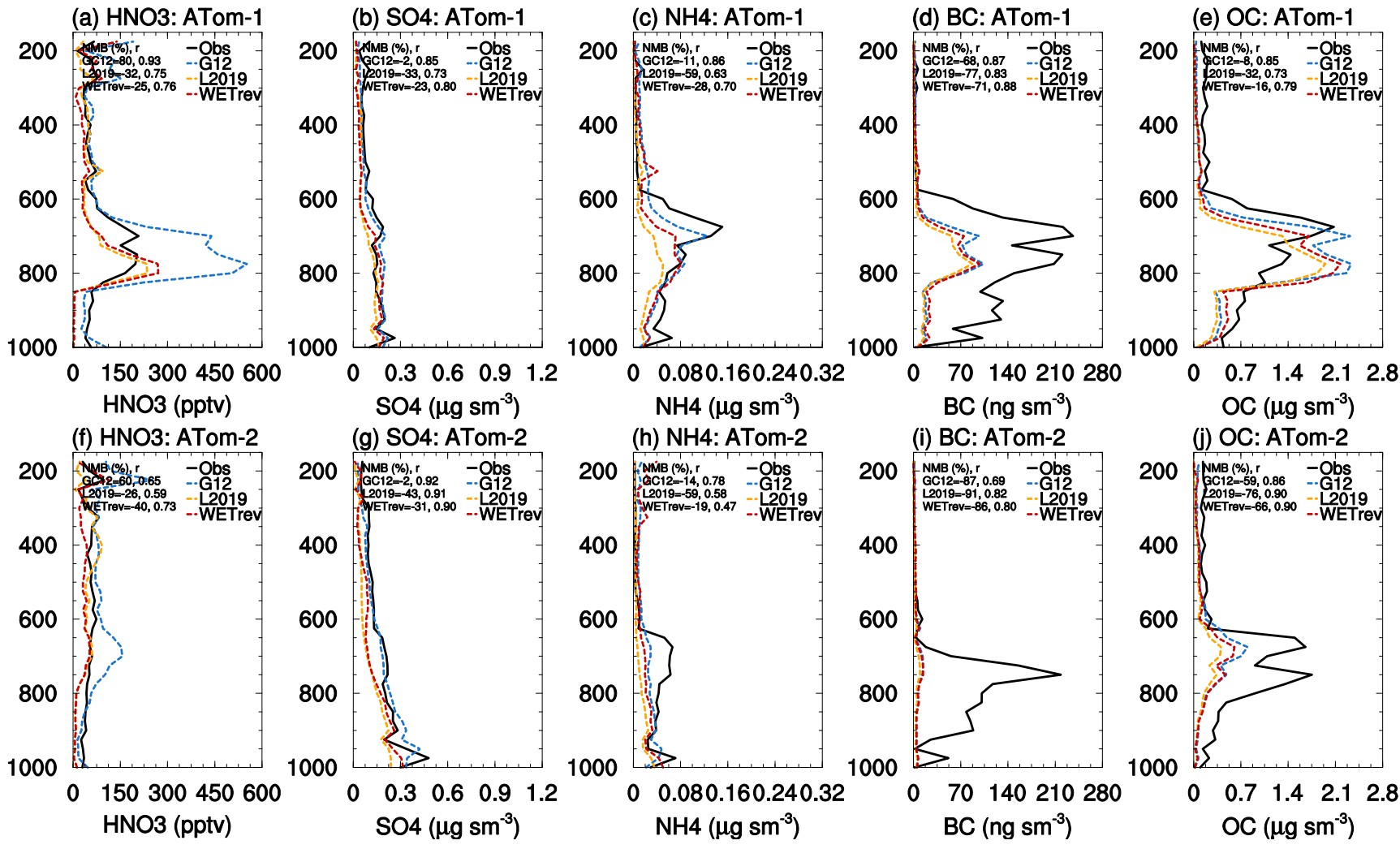

Figure 9. The same as Fig. 8 but over the Southern Hemisphere.

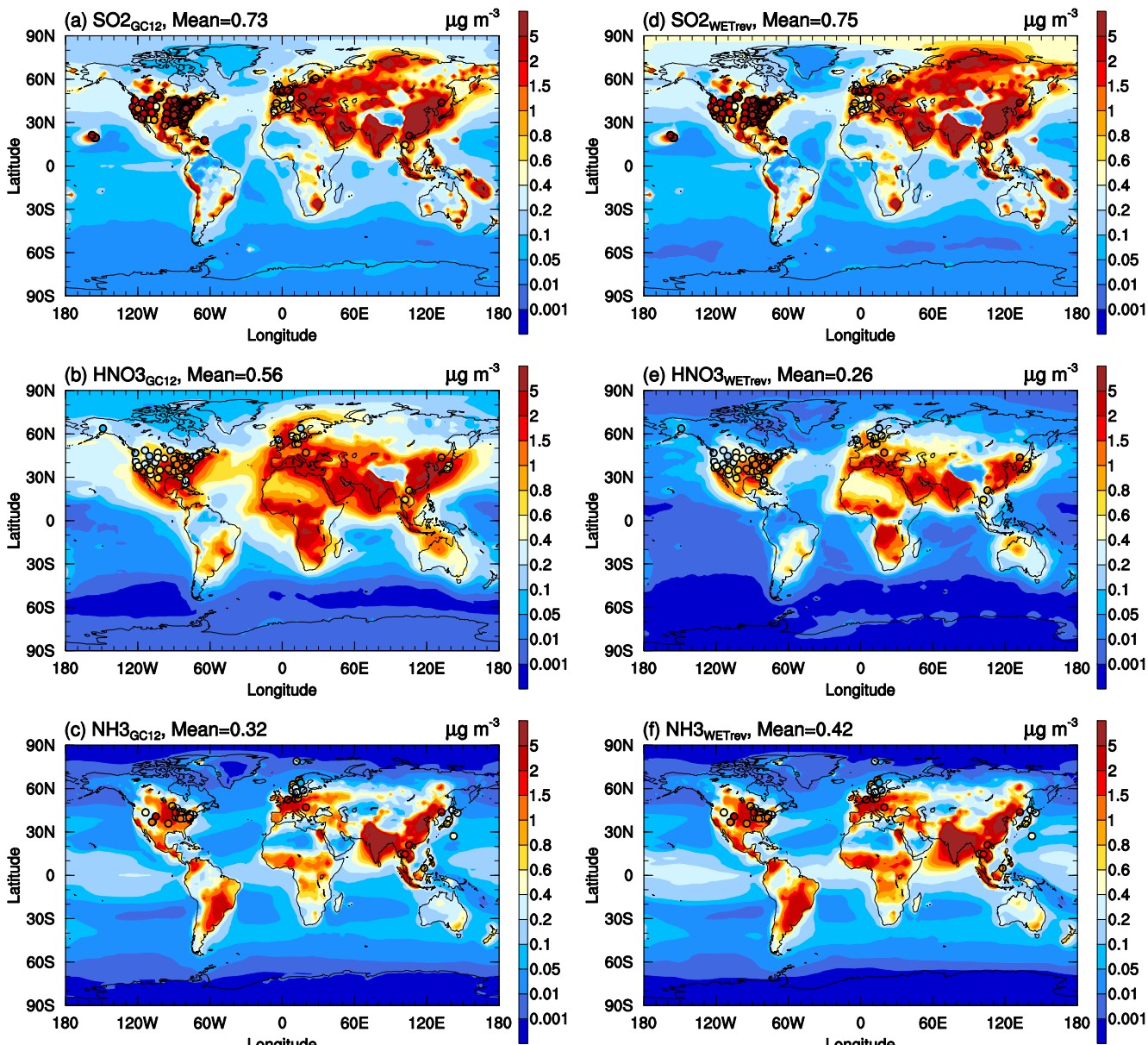

Figure 10. Horizontal distributions of SO₂, nitric acid, and ammonia surface mass concentrations simulated by (a-c) GC12 case and (d-f) WETrev case. Filled circles are annual mean surface mass concentrations observed at IMPROVE, CSN, CASTNET, AMoN, EMEP, and EANET for corresponding species.

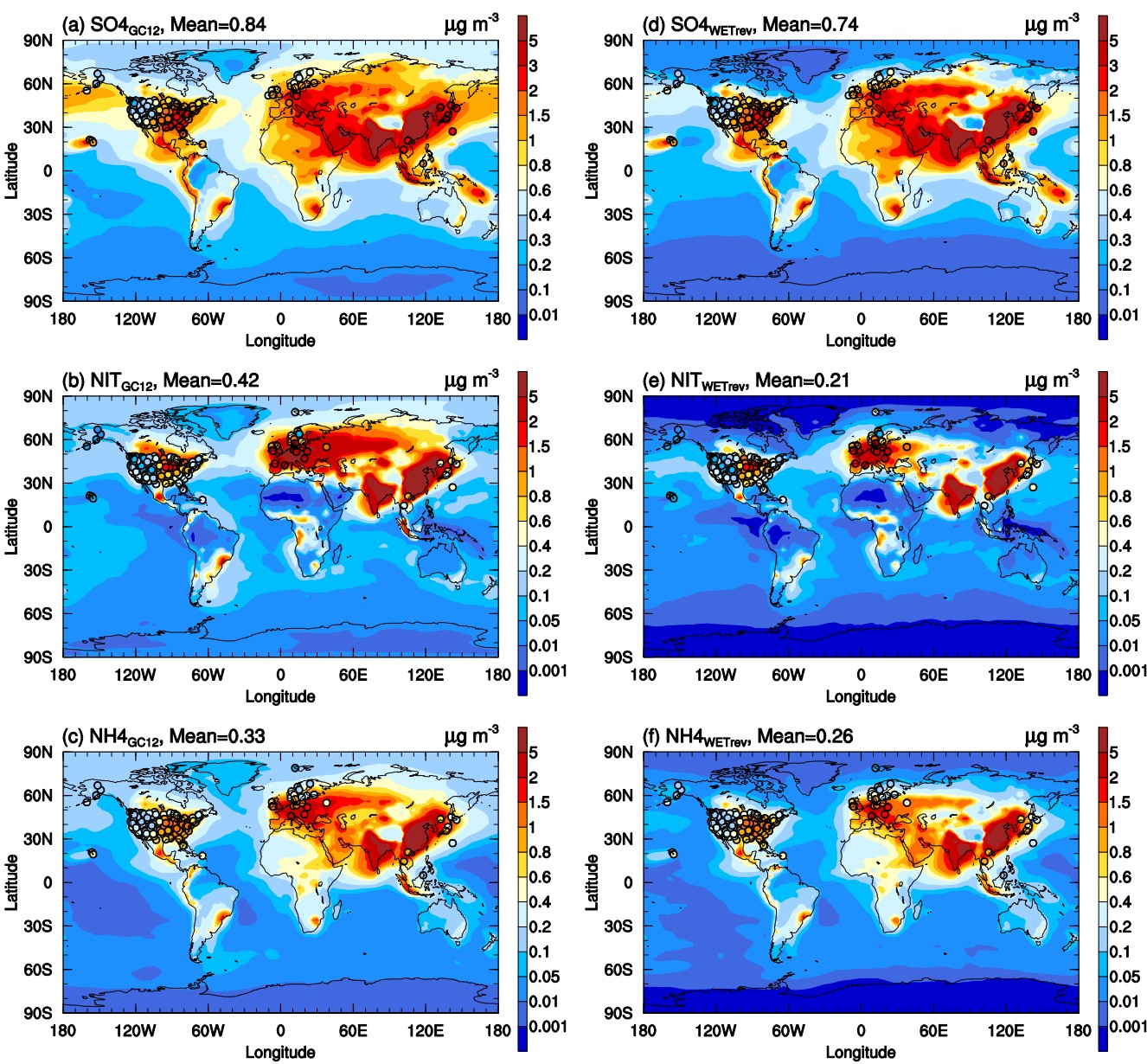

Figure 11. The same as Fig. 10 but for sulfate, nitrate, and ammonium surface mass concentrations.

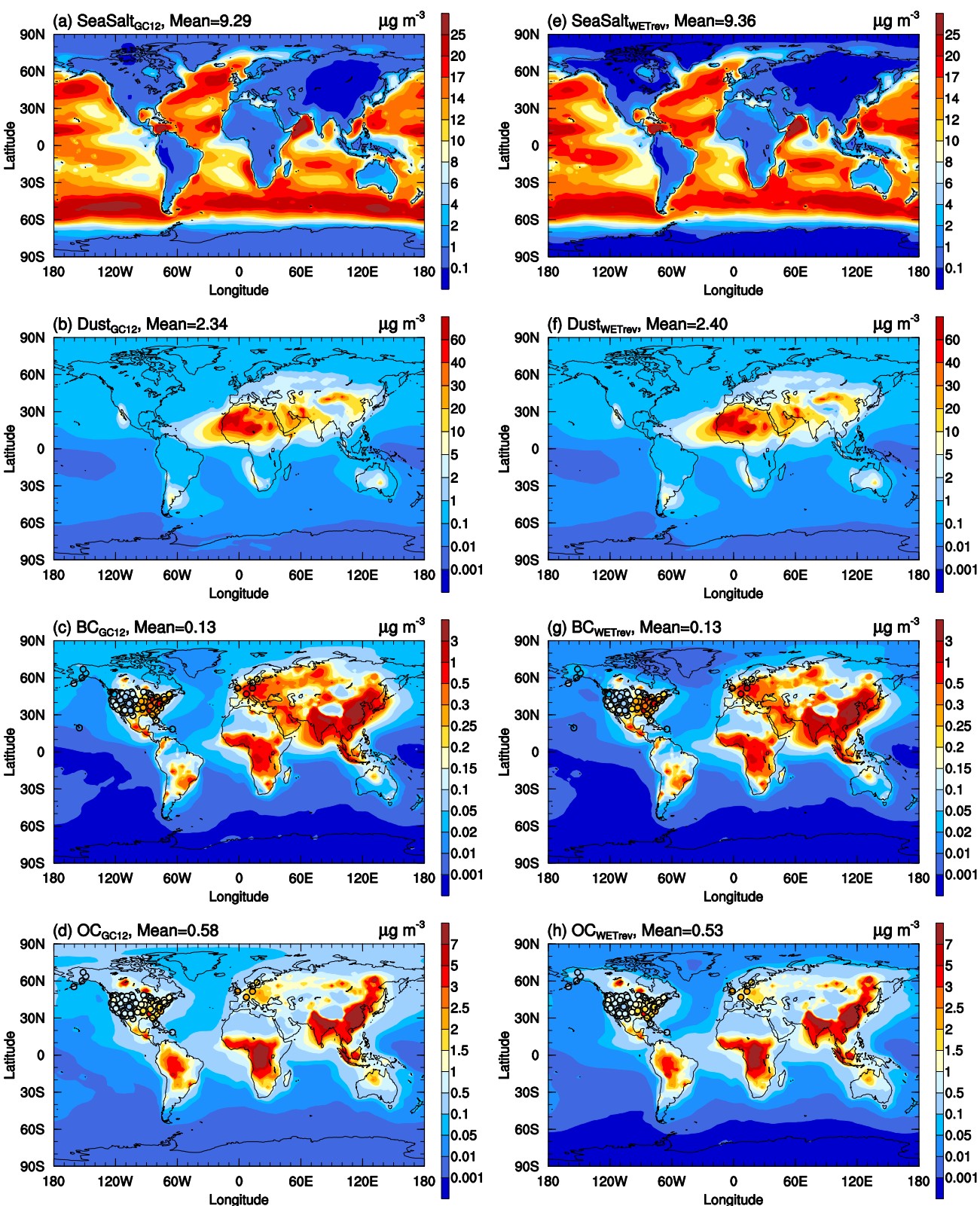

Figure 12. The same as Fig. 10 but for black carbon, organic carbon, sea salt, and dust surface
mass concentrations.

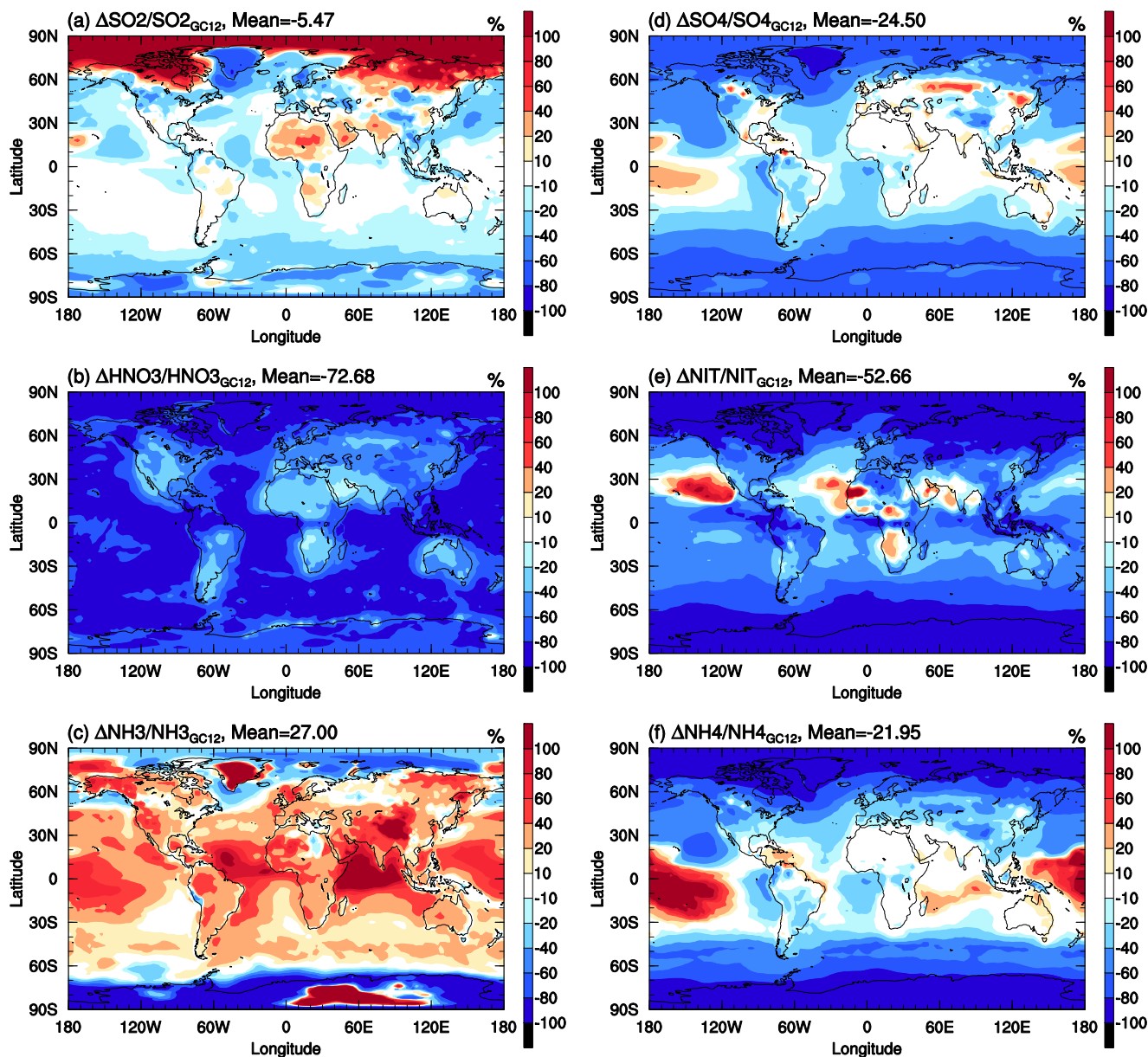

Figure 13. Horizontal distributions of percentage changes in annual mean (a) SO₂, (b) nitric acid, (c) ammonia, (d) sulfate, (e) nitrate, and (f) ammonium surface mass concentrations due to the switching of GC12 case to WETrev case.

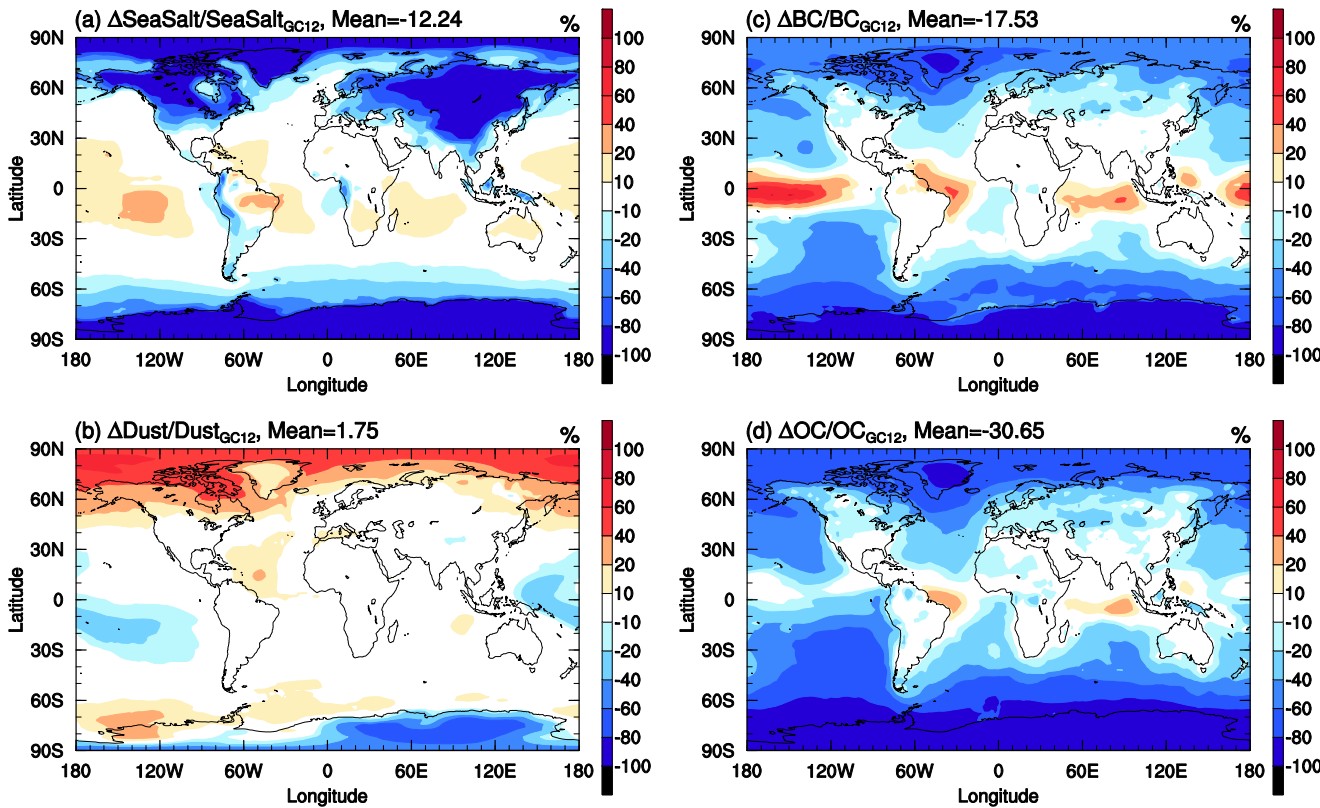

2 Figure 14. The same as Fig. 13 but for (a) sea salt, (b) dust, (c) black carbon, and (d) organic
3 carbon surface mass concentrations.