# Peer review of "Further improvement of wet process treatments in GEOS-Chem v12.6.0: Impact on global"

_Geoscientific Model Development, 2020_

## Referee Comment (RC1) · Anonymous Referee #1 · 28 Feb 2020

Review of "Further improvement of wet process treatments in GEOS-Chem v12.6.0: Impact on global distributions of aerosol precursors and aerosols"

**General comments**

The study attempts to improve the simulation of aerosol precursors and aerosols in GEOS-Chem via multiple updates in wet processes in the model. The updates in the treatment of wet processes have been described in details and the results are also evaluated with a large set of in-situ measurements from both surface monitoring networks and aircraft campaign. While the evaluation shows significant improvement in model results, it is not immediately clear how significant the update in each wet process actually is.

First, the updates in wet processes in this study including pH calculation for cloud, rain and wet surface, fraction of cloud available for aqueous phase chemistry, rainout efficiencies, washout efficiencies and wet surface uptakes during dry deposition. Evaluation of each update is necessary to understand the factors contributing to the uncertainties in the simulation of aerosol precursors and aerosols so that similar improvement could be applied to other models.

Second, the update in aqueous phase chemistry seems important for aerosol precursors and the corresponding aerosol species. But to what extent is the cloud/rain pH and subsequent dissolution in WETrev different from those in GC12?

In summary, this paper is well written and is easy to follow along. Its topic fits "Geoscientific Model Development" and it is worthy of publication subject to addressing these and specific comments below.

**Specific comments**

p. 6, line 27-30: what is the range of the calculated rainwater pH in this study?

p.11, line 3-4: In which way is ICCW related to wet scavenging? In other words, which equation is ICCW applied for?

p.12, line 2: why are the washout coefficients different between hydrophobic and hydrophilic aerosols? The mechanism associated with washout includes processes such as diffusion, interception, and impaction. Not sure how and to what extent it is affected by the water solubility of the aeosols.

p.15, line 19: it also enhances so2 at Zeppling in January and February, but not December, why?

p.16, line 17-21: I see the opposite way, where WETrev significantly underestimates nitric acid at the upper troposphere from Fig. 5

p. 34: reduce the xrange of the figure so that the difference among these lines can been seen more clearly

---

## Referee Comment (RC2) · Anonymous Referee #2 · 26 Mar 2020

Interactive comment on "Further improvement of wet process treatments in GEOS-Chem v12.6.0: Impact on global distributions of aerosol precursors and aerosols" The authors aim to improve the wet processes simulation of aerosols and aerosol precursors in GEOS-Chem v12.6.0 by further revising their previous work Luo et al. (2019), including updates to aqueous-phase chemistry and wet scavenging of aerosols and aerosol precursors in and below different types of cloud and during different types of precipitation, as well as dry deposition to different wet surface. The authors evaluate their updated wet process simulations using surface and aircraft measurements of aerosols and aerosol precursors concentrations from the US, Europe, Asia and Arctic, as well as AToM-1 and AToM-2 campaigns. This work is interesting and this topic is

important for regional and global modeling of aerosol and aerosol precursors.

However, major revision is recommended before being suitable for publication. While this work represents an admirable set of updates that are ostensibly improvements, the main drawback is that there is no systematic exploration of the impacts of any of the updates included in this paper. All changes are updated simultaneously, and then the model is evaluated in a rather generic fashion, without for example seeking out spatiotemporal subsets of data that would be most useful for isolating the impacts of any of the processes studied here. Notably, neither wet deposition measurements nor precipitation measurements are considered in the model evaluation. While the overall simulation updates are indeed an improvement, the paper leaves a bit to be desired in terms of explanations and scientific analysis. I believe addressing these requires more targeted use of the observations, additional simulation that examine the impact of subsets of the model updates tested individually, and evaluation of these to potentially refine some of the assumptions made during the model development. The work needs extensive proofreading throughout (every paragraph contains several grammatical errors; it goes beyond what I'm willing to edit myself), and several of the references are inappropriate. Further comments are described below.

Major comments: 1. As this work focus on improving the wet processes (mainly wet scavenging) simulation of aerosols and aerosol precursors, validation by measurements of wet deposition of aerosols and aerosol precursors is quite necessary given the availability of a bunch of wet deposition measurements over the US (http://nadp.slh.wisc.edu/ntn/ ), Europe (https://projects.nilu.no//ccc/emepdata.html ) and China (https://www.nature.com/articles/s41597-019-0061-2) and also from EANET network (https://www.eanet.asia/about/site-information/ ). Please at least validate your simulated wet deposition of sulfate + SO2, nitrate + HNO3, ammonium + ammonia using available measurements. Precipitation itself should also be evaluated.

2. Although the authors claim that they have surface measurement-based validation for Asia, the number of the Asian sites is very limited and none of these sites is in China or

India, where there are high emissions of aerosols and aerosol precursors and a large amount of precipitation. So robust validations in China and India are necessary if the authors claim that their updates also improve aerosol and aerosol precursor simulation for Asia.

3. The number of AMoN/EANET sites in Figure 7 seems to be much fewer than the total number of AMoN/EANET sites. Please give a brief description of all the measurements (IMPROVE, CSN, CASTNET, AMoM, EMEP, EANET, AToM-1 and AToM-2), number of valid sites, and data filtering you are using.

4. Quantitative evaluation for simulated vertical profiles of aerosol and aerosol precursors using AToM-1 and AToM-2 aircraft measurements is necessary to support your conclusions about improvements using your updated wet processes.

5. Section 2.6: How are wet surfaces defined in GEOS-Chem? Are these based on land-type or some other classification? Are they altered by precipitation? Overall this strikes me as a level of details beyond what this model can actually resolve.

6. Fig 1: a. It seems evident that the updates in both L2019 and WETrev degraded the model performance for SO2 and SO4 in the US, especially for SO2 This needs to be mentioned, explicitly, and discussed.

b. Why is the modeled SO4 seasonality incorrect in comparison to EANET?

c. Fig 1: Are model values for entire region or only sites at which observations are available?

d. Other factors mentioned in previous studies that may possibly impact overestimated HNO3 and nitrate concentrations are the constant hourly emissions of NH3. Has that been addressed here?

e. The authors seem to gloss over the impacts on NH3. First, the initial model performance compared to the observed NH3 concentrations is surprisingly good, given uncertainties in NH3 emissions. Second, there does seem to be significant difference

between WETrev and the other simulations, in comparison to the observations. In many months it would appear the bias compared to the measurements has increased by up to a factor of 2. None of the simulations correctly replicate the spring time NH3 maximum, most notably in Europe.

7. Figs 7 - 8: It is hard to get much out of the comparison to observations in these figures. Those are better represented by the previous figures, or require zoomed subplot of the US, Europe and E Asia. What would be more useful in Figs 7 - 8 would be to see the base case model (GC12) and the differences between this model and WETrev, as absolute and relative differences.

Minor comments:

1. Page 1, line 25-28: "we compared model simulation . . . successfully improved by considering the updated wet processes." Please give quantitative metrics to support this conclusion.

2. Pag1, line 20: So results for the decrease in NMB of these species in the US are from L2019? If so, probably should't be presented in the abstract as results from the current study.

3. Page 3, line 26 and page 12, line16: I don't think the web site wiki is a suitable citation. Please refer the peer reviewed literature upon which such material is based.

4. Page 6, line 16-17: "we assume that total amount . . .aerosol thermodynamics (SNVC) is 25 % of sulfate". Why do you use 25% here? You should have some rational for this 25% although it is assumed.

5. Page, line 22-30: You are encouraged to validate your rainwater pH using precipitation pH measurements over the US, Europe and Asia.

6. Page 6, line 2: I'm curious how this problem is formulated and how the updated solution method using Newton's method is applied. These are the sorts of details that should be described explicitly here, at the level at which they are reproducible

from reading the text. Citation of unpublished preliminary work from a conference presentation (Moch 2019) is not an adequate reference nor explanation.

7. Page 6, line 17: What is the basis for picking 25% here? No explanation or reference has been provided.

8. Section 2: It's not clear how this particular model distinguishes / defines the processes of rainout vs washout — could this be clarified?

9. Page 7, line 22-27: I can understand that you take the ice surface as an aqueous layer when temperature higher than 263K in the mixed cloud, and this means aqueous chemistry can happen at the surface layer of the ice cloud. But I don't understand why it is reasonable to assume aqueous phase cloud fraction equals grid mean cloud fraction: faq=fc. This formula means you assume the aqueous chemistry happen in the ice cloud the way as it does in an aqueous cloud, not just at the pre-melt layer of the ice cloud, most of which is ice-phase. The ice-phase cloud water in one grid is 3 dimensional and I can accept the assumption that the aqueous chemistry can happen at the surface pre-melt layer, but it doesn't make sense that the aqueous chemistry can happen in an ice-phase cloud the way in aqueous-phase cloud. Also, does this assumption of aqueous layer of ice also increase the wet scavenging of HNO3 in the same grid cell?

10. Page 8, line 14 – 18: It's not clear to me this is double counting. It seems one process describes the absorption and the other the oxidation, with both steps being necessary. Am I missing something here? If so, could the authors explain the model treatment of these processes in more detail? Schematics could be helpful.

11. Page 8, line 25-30: "the rationale . . . as water soluble aerosols . . .. The composition . . . for cloud activation calculation." Please include some appropriate references here.

12. Page 9, line 1-3: "However, in the actual atmosphere, . . . coated with SNA". Please also include some appropriate references here.

13. Page 9, line 29-30: "While most of . . .smaller than 500 nm." Please include some appropriate references here.

14. Page 10, line 4 – 8: How was this treated in the original model?

15. Page 13, line 15-17: "One possible reason is . . . at urban sites . . .remote regions." Please include some appropriate references here.

16. Page 13, line 29: Is there a scientific explanation why one would revert this change, in terms of understanding of heterogeneous chemistry? Or are the authors alternatively suggesting that these parameters be adjusted so that the model estimates better fit the data?

17. Page 14: line 1-2: "The aqueous concentration of ammonia is much lower than nitric acid, . . . small impact on the simulation of ammonia." How does this conclusion come? Isn't it because the increasing ammonia wet deposition is compensated by less reaction with decreased HNO3 in the air?

18. Page 14, line 27-28: "The underestimate is likely . . . wildfire . . . US." Please include some appropriate references here.

19. Page 15, line 30-31: "Most of BC at Arctic . . . anthropogenic emissions," Please include some appropriate references here.

20. Page 16, line 7-11: Please provide figures for the tracks of ATom-1 and ATom-2. And why do you filter out tracks over land? Could you provide profile comparison over land?

21. For the whole section 3.3, please give quantitative metrics (e. g. NMB and R) to show how your updated wet scavenge schemes work in reproducing the observed profiles of these chemical species. Figure 5 (a) shows WETrev underestimate HNO3 throughout the whole layers except for high bias at upper layer (nearby 200 hpa). Figure 6 shows that L2019 and WETrev largely underestimate HNO3 at lower troposphere (> 800 hpa).

22. Page 17, line 6: With regards to "in this work," it seems that a bulk of the improvements are from the updates in L2019, more so than the present work.

23. Page 18 line 25 - page 19 line 6: please try to quantitively and appropriately show how your updates improve the simulation. Like your 5th conclusion on page 19 line 5-6: "The updated wet surface ... SO2 at Artic sites", I am not so sure whether your conclusion is fully supported by figure 3 or not.

---

## Referee Comment (RC3) · Anonymous Referee #3 · 27 Mar 2020

This paper presented updated treatments of wet processes in GEOS-Chem, including rainout efficiencies for warm, mixed-phase and cold clouds, empirical washout by rain/snow, aqueous phase chemistry and wet removal for SO2 and sulfate, and wet surface uptakes during dry deposition. Model simulated concentrations of aerosols and aerosol precursors were evaluated with various surface observational data sets over the U.S., Europe, Asia, and Arctic as well as aircraft measurements of nitric acid and aerosols during two ATom campaigns. Results showed significant improvement over previous version of the model and better agree with the observations. Although mentioned in various places in the paper, the roles of individual wet processes in the improvements were not systematically quantified. This paper is well organized and

overall well written, but needs careful proofreading. I recommend publication after the following comments are addressed.

P4, Line 3, eqn 1: Pr is the grid-box large-scale precipitation (rain+snow) formation rate. LCW is liquid phase cloud water content. But the total condensed water content should also include ice cloud water content, which is missing from this equation.

P8, Line 7, eqn 11: same issue as for eqn 1. For T>=258K (warm clouds), this equation assumes zero ice cloud water (ICW), which is probably not true in MERRA-2. Since the model uses temperature ranges to separate scavenging due to warm/mixed-phase/cold clouds, the cloud condensed water (for all T) needs to include ICW. This is expected to have a significant impact on the model results of this paper.

P10-11, Section 2.4: For rainout in cold cloud (T<237K), do you limit it to below the MERRA-2 tropopause?

P16, L17-19, and Fig.5: Please double check. It looks like the L2019 and WETrev lines for HNO3 are switched. What aspect of the "old treatments in GC12" do you mean here?

P19, Code and data availability: the revised GEOS-Chem v12.6.0 code and model output need to be made available at a public data depository. Also it's not clear where the various observational data sets used in this work were downloaded from.

Minor comments:

P4, Line 28: is LW different than LCW in eqn 1.

P5, L1 (and other places): Do you mean "acidity"?

P5, L3: H* can be calculated . . .

P5, L8-9: what are the units for these constants and coefficients?

P6, L19: the comma is misplaced.

P7, L20: LCW not LWC

P13, L3: Emissions are produced by the default setting of HEMCO. Does this mean that emissions are specific to the periods of ATom-1 and ATom-2 campaigns?

P13, L16: Is there a reference for "a large amount of USEPA observations are located at urban regions"?

P14, L20: remove "– ".

P15, L7: low dissolution

P15, L27 & L29: "at Alert during spring" – during winter / early spring?

P16, L1: converted

P16, L10-11: Why are the flight tracks over the land filtered out for comparison?

P18, L18-24: this sentence needs a break.

P19, L3, L12: remove"an"; exist.

Table 1: refer the reader to eqn 16.

Proofread and check for grammatical errors.

---

## Author Comment (AC1) · 24 Apr 2020

We thank the referee for the detailed reviews and constructive comments that help to improve the manuscript. Below we respond to the comments in detail. (Referee's comments are in Italic).

*General comments The study attempts to improve the simulation of aerosol precursors and aerosols in GEOSChem via multiple updates in wet processes in the model. The updates in the treatment of wet processes have been described in details and the results are also evaluated with a large set of in-situ measurements from both surface monitoring networks and aircraft campaign. While the evaluation shows significant*

[Figure]

*improvement in model results, it is not immediately clear how significant the update in each wet process actually is. First, the updates in wet processes in this study including pH calculation for cloud, rain and wet surface, fraction of cloud available for aqueous phase chemistry, rainout efficiencies, washout efficiencies and wet surface uptakes during dry deposition. Evaluation of each update is necessary to understand the factors contributing to the uncertainties in the simulation of aerosol precursors and aerosols so that similar improvement could be applied to other models.*

Because of the lack of detailed process diagnostics in GEOS-Chem, it is difficult to trace the contributions from each modifications. To address the referee's comment, we carried out 5 numerical sensitivity study cases (RO, WO, RP, DD, and AC) to understand the factors contributing to the uncertainties in the simulation of aerosol precursors and aerosols. RO case is the same as case WETrev except using rainout rate in GC12; WO case is the same as case WETrev except using washout rate in GC12; RP case is the same as case WETrev except assuming pH of rainwater for wet scavenging is 4.5; DD case is the same as case WETrev except using dry deposition treatment in GC12; and AC case is the same as case WETrev except using aqueous phase chemistry treatment in GC12. The results of the sensitivity study and associated discussions have been added to section 3.1.

*Second, the update in aqueous phase chemistry seems important for aerosol precursors and the corresponding aerosol species. But to what extent is the cloud/rain pH and subsequent dissolution in WETrev different from those in GC12?*

In GC12, pHs of cloud and rain for wet deposition are assumed to be 4.5. The calculated rainwater pH in this study varied from 4.3 to 6.9. The impact of pH on effective Henry's law constant of $SO_2$ is shown in Table 1. We have clarified this in the revised text.

*Specific comments p. 6, line 27-30: what is the range of the calculated rainwater pH in this study?*

The calculated rainwater pH in this study varied from 4.3 to 6.9. This is clarified in the text.

*p.11, line 3-4: In which way is ICCW related to wet scavenging? In other words, which equation is ICCW applied for?*

ICCW is applied for equation 1. This is clarified in equation and the text.

*p.12, line 2: why are the washout coefficients different between hydrophobic and hydrophilic aerosols? The mechanism associated with washout includes processes such as diffusion, interception, and impaction. Not sure how and to what extent it is affected by the water solubility of the aeosols.*

The assumption of different washout coefficients for hydrophobic and hydrophilic aerosols is because the rain washout rate for water-soluble aerosols measured by Laakso et al. (2003) is still 20 times larger than that calculated by the semi-empirical parameterization. One of the possible reasons is droplet–particle collection mechanisms for hydrophobic and hydrophilic aerosols are different. This is clarified in the text.

*p.15, line 19: it also enhances so2 at Zeppling in January and February, but not December, why?*

At Zeppelin, temperature in December is higher than that in January and February. SO2 is enhanced due to the modification of dry deposition in this work. However, there is more aqueous phase chemistry in December which consumes the enhanced SO2. This is clarified in the text.

*p.16, line 17-21: I see the opposite way, where WETrev significantly underestimates nitric acid at the upper troposphere from Fig. 5*

For nitric acid above 300 hPa, the values simulated by WETrev are higher than those by L2019. For nitric acid between 500 hPa and 300 hPa, the values simulated by WETrev are lower than those by L2019. It is because L2019 only considered washout of nitric

acid by rain. WETrev also considered washouts of nitric acid by snow and ice which were absent in L2019 and GC12. This is clarified in the text.

*p. 34: reduce the xrange of the figure so that the difference among these lines can been seen more clearly*

X-ranges of Figure 5 and Figure 6 were determined by maximum values of each species during ATom-1 and ATom-2. It lets readers easily find the spatial and temporal variations of these species at the North Hemisphere and South Hemisphere during boreal summer (ATom-1) and boreal winter (ATom-1). We keep the original x-range as we think it is more suitable for what we want to present.
* * *
|        | 273K     | 283K     | 293K    | 303K    |
|--------|----------|----------|---------|---------|
| pH=4.5 | 2479.7   | 1271.7   | 682.7   | 382.0   |
| pH=5   | 7888.3   | 4039.5   | 2165.9  | 1210.4  |
| pH=5.5 | 25474.2  | 12995.7  | 6947.4  | 3873.3  |
| pH=6   | 85913.1  | 43355.2  | 22980.1 | 12724.6 |
| pH=6.5 | 325309.2 | 159736.0 | 82801.8 | 45021.4 |

**Fig. 1.** Table 1. The values of effective Henry's law constant of SO2.

---

## Author Comment (AC2) · 24 Apr 2020

We thank the referee for the detailed reviews and constructive comments that help to improve the manuscript. Below we respond to the comments in detail. (Referee's comments are in Italic).
*aerosols and aerosol precursors in and below different types of cloud and during different types of precipitation, as well as dry deposition to different wet surface. The authors evaluate their updated wet process simulations using surface and aircraft measurements of aerosols and aerosol precursors concentrations from the US, Europe, Asia and Arctic, as well as AToM-1 and AToM-2 campaigns. This work is interesting and this topic is important for regional and global modeling of aerosol and aerosol precursors.*

Thanks for the positive comment about the importance of this work.

*However, major revision is recommended before being suitable for publication. While this work represents an admirable set of updates that are ostensibly improvements, the main drawback is that there is no systematic exploration of the impacts of any of the updates included in this paper. All changes are updated simultaneously, and then the model is evaluated in a rather generic fashion, without for example seeking out spatiotemporal subsets of data that would be most useful for isolating the impacts of any of the processes studied here. Notably, neither wet deposition measurements nor precipitation measurements are considered in the model evaluation. While the overall simulation updates are indeed an improvement, the paper leaves a bit to be desired in terms of explanations and scientific analysis. I believe addressing these requires more targeted use of the observations, additional simulation that examine the impact of subsets of the model updates tested individually, and evaluation of these to potentially refine some of the assumptions made during the model development. The work needs extensive proofreading throughout (every paragraph contains several grammatical errors; it goes beyond what I'm willing to edit myself), and several of the references are inappropriate. Further comments are described below.*

Thanks for the constructive comments. The manuscript has been revised following the suggestions.

*Major comments: 1. As this work focus on improving the wet processes (mainly wet scavenging) simulation of aerosols and aerosol precursors, validation by mea-*

*surements of wet deposition of aerosols and aerosol precursors is quite neces-
sary given the availability of a bunch of wet deposition measurements over the US
(http://nadp.slh.wisc.edu/ntn/ ), Europe (https://projects.nilu.no//ccc/emepdata.html )
and China (https://www.nature.com/articles/s41597-019-0061-2) and also from EANET
network (https://www.eanet.asia/about/site-information/ ). Please at least validate your
simulated wet deposition of sulfate + SO2, nitrate + HNO3, ammonium + ammonia
using available measurements. Precipitation itself should also be evaluated.*

We added comparison of simulated wet deposition of sulfate+SO2, nitrate+HNO3, am-
monium+NH3 with available measurements. Associated discussions are in section
3.1.

*2. Although the authors claim that they have surface measurement-based validation for
Asia, the number of the Asian sites is very limited and none of these sites is in China or
India, where there are high emissions of aerosols and aerosol precursors and a large
amount of precipitation. So robust validations in China and India are necessary if the
authors claim that their updates also improve aerosol and aerosol precursor simulation
for Asia.*

We clarified in the revised paper that wet process updates improve aerosol and aerosol
precursor simulation over Asia remote regions.

*3. The number of AMoN/EANET sites in Figure 7 seems to be much fewer than the total
number of AMoN/EANET sites. Please give a brief description of all the measurements
(IMPROVE, CSN, CASTNET, AMoM, EMEP, EANET, AToM-1 and AToM-2), number of
valid sites, and data filtering you are using.*

The criterion of observations used for model validation is that valid data are available
for every month in 2011. For EANET observations, due to too much missing data,
the criterion is loosen to monthly mean data available for each month during a 3-year
period (2010-2012). Seto et al. (2007) pointed out that EANET observations at urban
sites are much higher than those at remote sites. Since the number of the Asian sites

is very limited, to make the validation more appropriate, only remote and rural sites are used for model validation. A brief description on these has been added at section 3.1.

Seto, S., Sato, M., Tatano, T., Kusakari, T. and Hara, H., Spatial distribution and source identification of wet deposition at remote EANET sites in Japan. Atmos. Environ.41,9386ËŮ9396, 2007.

*4. Quantitative evaluation for simulated vertical profiles of aerosol and aerosol precursors using AToM-1 and AToM-2 aircraft measurements is necessary to support your conclusions about improvements using your updated wet processes.*

Normalized mean biases (NMB) NMB and correlation coefficient (r) have been used to quantitative evaluation for simulated vertical profiles of aerosol and aerosol precursors with AToM-1 and AToM-2 aircraft measurements.

*5. Section 2.6: How are wet surfaces defined in GEOS-Chem? Are these based on land-type or some other classification? Are they altered by precipitation? Overall this strikes me as a level of details beyond what this model can actually resolve.*

GEOS-Chem determined wet surfaces based on land use type from the Olson 2001 land map (Olson, 1992). They are not altered by precipitation. This has been clarified in the text.

Olson, J, World Ecosystems (WE1.4): Digital raster data on a 10 minute geographic 1080 x 2160 grid, in Global Ecosystems Database, version 1.0, Disc A, edited by NOAA Natl. Geophys. Data Center, Boulder, Colorado, 1992.

*6. Fig 1: a. It seems evident that the updates in both L2019 and WETrev degraded the model performance for SO2 and SO4 in the US, especially for SO2 This needs to be mentioned, explicitly, and discussed.*

Accepted.

*b. Why is the modeled SO4 seasonality incorrect in comparison to EANET?*

It is caused by the overestimation of January SO4 at Primorskaya, Russia (43.63°N, 132.24°E) whose value is high up to 12 $\mu$g m-3, 2.5 times higher than observation at this site. This overestimation is associated with aqueous phase chemistry over there.

*c. Fig 1: Are model values for entire region or only sites at which observations are available?*

Model values were sampled at sites where observations are available.

*d. Other factors mentioned in previous studies that may possibly impact overestimated HNO3 and nitrate concentrations are the constant hourly emissions of NH3. Has that been addressed here?*

The present version of GEOS-Chem considers diurnal, seasonal, and interannual variability of ammonia emission.

*e. The authors seem to gloss over the impacts on NH3. First, the initial model performance compared to the observed NH3 concentrations is surprisingly good, given uncertainties in NH3 emissions. Second, there does seem to be significant difference between WETrev and the other simulations, in comparison to the observations. In many months it would appear the bias compared to the measurements has increased by up to a factor of 2. None of the simulations correctly replicate the spring time NH3 maximum, most notably in Europe.*

The unreasonable seasonal variation of ammonia in the Europe is caused by the updated emission treatment in GC12.6. GC12.6 replaced old EMEP emissions and seasonal scaling factors with CEDS global emissions. After switching back to EMEP emissions, seasonal variation of ammonia was captured by the model. We rerun the cases with EMEP emissions and updated the results in the revised manuscript.

*7. Figs 7 - 8: It is hard to get much out of the comparison to observations in these figures. Those are better represented by the previous figures, or require zoomed subplot of the US, Europe and E Asia. What would be more useful in Figs 7 - 8 would be to*

*see the base case model (GC12) and the differences between this model and WETrev, as absolute and relative differences.*

The two figures were provided to show the impacts of WETrev on a global scale. One can derive globally averaged absolute and relative differences from values given on top of each panel. Interested readers can use the enlarge function for pdf figures to zoom into specific regions. There will be too many figures if we provide zoomed subplots and plots for both absolute and relative differences.

*Minor comments: 1. Page 1, line 25-28: "we compared model simulation . . . successfully improved by considering the updated wet processes." Please give quantitative metrics to support this conclusion.*

Quantitative metrics have been added.

*2. Pag1, line 20: So results for the decrease in NMB of these species in the US are from L2019? If so, probably should't be presented in the abstract as results from the current study.*

The decrease in NMB of these species is mainly caused by the updated ICCW and empirical washout. Excluding the results in the US sounds strange here.

*3. Page 3, line 26 and page 12, line16: I don't think the web site wiki is a suitable citation Please refer the peer reviewed literature upon which such material is based.*

As shown in the web site wiki, H* of SO2, H2O2, and NH3 for dry deposition was originally in drydep$_mod.F.Wewerenotabletolocatethepeerreviewedliteratureonthis.$

*4. Page 6, line 16-17: "we assume that total amount . . .aerosol thermodynamics (SNVC) is 25 % of sulfate". Why do you use 25% here? You should have some rational for this 25% although it is assumed.*

It is based on the work of Guo et al. (2018) cited in the text which suggested ammonium-sulfate aerosol molar ratio is $1.47\pm0.43(\approx 1.5)$.

*5. Page, line 22-30: You are encouraged to validate your rainwater pH using precipitation pH measurements over the US, Europe and Asia.*

We will investigate this in our future work.

*6. Page 6, line 2: I'm curious how this problem is formulated and how the updated solution method using Newton's method is applied. These are the sorts of details that should be described explicitly here, at the level at which they are reproducible from reading the text. Citation of unpublished preliminary work from a conference presentation (Moch 2019) is not an adequate reference nor explanation.*

More detailed description on this has been added.

*7. Page 6, line 17: What is the basis for picking 25% here? No explanation or reference has been provided.*

The assumption of 25 % is based on the work of Guo et al. (2018) cited in the text. More robust calculation of SNVC need to be investigated in future works.

*8. Section 2: It's not clear how this particular model distinguishes / defines the processes of rainout vs washout could this be clarified?*

Rainout is the removal due to formation of precipitation in cloud, while washout is the removal due to falling precipitation from upper layers. This has been clarified in the text.

*9. Page 7, line 22-27: I can understand that you take the ice surface as an aqueous layer when temperature higher than 263K in the mixed cloud, and this means aqueous chemistry can happen at the surface layer of the ice cloud. But I don't understand why it is reasonable to assume aqueous phase cloud fraction equals grid mean cloud fraction: faq=fc. This formula means you assume the aqueous chemistry happen in the ice cloud the way as it does in an aqueous cloud, not just at the pre-melt layer of the ice cloud, most of which is ice-phase. The ice-phase cloud water in one grid is 3 dimensional and I can accept the assumption that the aqueous chemistry can happen*

[Figure]

*at the surface pre-melt layer, but it doesn't make sense that the aqueous chemistry can happen in an ice-phase cloud the way in aqueous-phase cloud. Also, does this assumption of aqueous layer of ice also increase the wet scavenging of HNO3 in the same grid cell?*

Yes, it is right. Due to uncertainty of the thickness of aqueous layer of ice, we decide to only use equation 9 to calculate aqueous phase cloud fraction. The assumption of aqueous layer of ice does not increase the wet scavenging of HNO3 in the same grid cell.

*10. Page 8, line 14 – 18: It's not clear to me this is double counting. It seems one process describes the absorption and the other the oxidation, with both steps being necessary. Am I missing something here? If so, could the authors explain the model treatment of these processes in more detail? Schematics could be helpful.*

In GC12, rainout of SO2 is limited by the aqueous phase oxidation of SO2 by H2O2 rather than the absorption by cloud water (Chin et al., 1996). However, the conversion of SO2 to sulfate in cloud has been accounted for in the aqueous phase chemistry. In this work, rainout of SO2 is limited by the absorption by cloud water. More explanation of the processes has been added in the revised text.

*11. Page 8, line 25-30: "the rationale . . . as water soluble aerosols . . .. The composition . . . for cloud activation calculation." Please include some appropriate references here.*

Added. Abdul-Razzak, H., and Ghan, S. J., A parameterization of aerosol activation: 2. Multiple aerosol types, J. Geophys. Res., 105( D5), 6837– 6844, doi:10.1029/1999JD901161, 2000.

*12. Page 9, line 1-3: "However, in the actual atmosphere, . . . coated with SNA". Please also include some appropriate references here.*

Added. Fassi-Fihri, A., Suhre, K., and Rosset, R.: Internal and external mixing in

atmospheric aerosols by coagulation: impact on the optical and hygroscopic properties of the sulphate-soot system, Atmos. Environ., 10, 1393–1402, 1997.

*13. Page 9, line 29-30: "While most of . . .smaller than 500 nm." Please include some appropriate references here.*

Added. Sahu, L. K., Y. Kondo, N. Moteki, N. Takegawa, Y. Zhao, M. J. Cubison, J. L. Jimenez, S. Vay, G. S. Diskin, A. Wisthaler, T. Mikoviny, L. G. Huey, A. J. Weinheimer, D. J. Knapp, Emission characteristics of black carbon in anthropogenic and biomass burning plumes over California during ARCTAS‐CARB 2008, J. Geophys. Res., 117, D16302, doi:10.1029/2011JD017401, 2012.

Zender, C. S., Bian, H., and Newman, D., Mineral Dust Entrainment and Deposition (DEAD) model: Description and 1990s dust climatology, J. Geophys. Res., 108, 4416, doi:10.1029/2002JD002775, D14, 2003.

*14. Page 10, line 4 – 8: How was this treated in the original model?*

GC12 assumed rainout efficiency of water-soluble aerosols by cold cloud is 100 %. This is clarified in the revised text.

*15. Page 13, line 15-17: "One possible reason is . . . at urban sites . . .remote regions." Please include some appropriate references here.*

We didn't find related reference. There were 288 EPA's Air Quality System sites with valid data in each month of 2011. 69 of these sites were with the mark of 'Not in a city'. More information can be found at https://www.epa.gov/outdoor-air-quality-data.

*16. Page 13, line 29: Is there a scientific explanation why one would revert this change, in terms of understanding of heterogeneous chemistry? Or are the authors alternatively suggesting that these parameters be adjusted so that the model estimates better fit the data?*

Uptake coefficients for heterogeneous chemistry on sulfate in the work of Holmes et

al. (2019) are 10 times smaller than those used in GC12.5 which lead to less nitric acid production in GC12.6 than that in GC12.5. Due to large uncertainties of uptake coefficients for heterogeneous chemistry, further investigations are needed. Yes, at this point, we are suggesting that these parameters be adjusted so that the model estimates better fit the data.

*17. Page 14: line 1-2: "The aqueous concentration of ammonia is much lower than nitric acid, . . . small impact on the simulation of ammonia." How does this conclusion come? Isn't it because the increasing ammonia wet deposition is compensated by less reaction with decreased HNO3 in the air?*

This is a good point and we agree it is because the increasing ammonia wet deposition is compensated by less reaction with decreased HNO3 in the air. We have modified the sentence to reflect this.

*18. Page 14, line 27-28: "The underestimate is likely . . . wildfire . . . US." Please include some appropriate references here.*

Added. Mao, Y. H., Li, Q. B., Henze, D. K., Jiang, Z., Jones, D. B. A., Kopacz, M., He, C., Qi, L., Gao, M., Hao, W.-M., and Liou, K.-N.: Estimates of black carbon emissions in the western United States using the GEOS-Chem adjoint model, Atmos. Chem. Phys., 15, 7685–7702, https://doi.org/10.5194/acp-15-7685-2015, 2015.

*19. Page 15, line 30-31: "Most of BC at Arctic . . . anthropogenic emissions," Please include some appropriate references here.*

Added. Xu, J.-W., Martin, R. V., Morrow, A., Sharma, S., Huang, L., Leaitch, W. R., Burkart, J., Schulz, H., Zanatta, M., Willis, M. D., Henze, D. K., Lee, C. J., Herber, A. B., and Abbatt, J. P. D.: Source attribution of Arctic black carbon constrained by aircraft and surface measurements, Atmos. Chem. Phys., 17, 11971–11989, https://doi.org/10.5194/acp-17-11971-2017, 2017.

*20. Page 16, line 7-11: Please provide figures for the tracks of ATom-1 and ATom-2.*

[Figure]

*And why do you filter out tracks over land? Could you provide profile comparison over land?*

The flight tracks of ATom-1 and ATom-2 are shown in Fig. 1, while vertical profiles over land are shown in Fig. 2. These figures have been provided in supplementary materials. ATom observations over the land, whose values vary greatly, only account for 28 % of total measurements. To make the comparison more appropriate, we filtered out the flight tracks over the land.

*21. For the whole section 3.3, please give quantitative metrics (e. g. NMB and R) to show how your updated wet scavenge schemes work in reproducing the observed profiles of these chemical species. Figure 5 (a) shows WETrev underestimate HNO3 throughout the whole layers except for high bias at upper layer (nearby 200 hpa). Figure 6 shows that L2019 and WETrev largely underestimate HNO3 at lower troposphere (>800 hpa).*

Added as suggested.

*22. Page 17, line 6: With regards to "in this work," it seems that a bulk of the improvements are from the updates in L2019, more so than the present work.*

We modified it as: the updated wet process treatments in this work and L2019 can improve the agreements of simulated and observed vertical profiles of nitric acid and aerosols.

*23. Page 18 line 25 - page 19 line 6: please try to quantitively and appropriately show how your updates improve the simulation. Like your 5th conclusion on page 19 line 5-6: "The updated wet surface . . . SO2 at Artic sites", I am not so sure whether your conclusion is fully supported by figure 3 or not.*

Accepted. Quantitative metrics have been added. For 5th conclusion, we found NMB of SO2 is increased from -23 % to 32 % at Nord and decreased from 27 % to 2(5) The updated wet surface uptake during dry deposition changes the performance of

[Figure]

simulated SO2 at Arctic sites. NMB of SO2 is increased from -23 % to 32 % at Nord and decreased from 27 % to 22 % at Zeppelin.

[Figure]

[Figure]

**Fig. 1.** The flight tracks of (a) ATom-1 and (b) ATom-2.

[Figure]

**Fig. 2.** Vertical profiles of nitric acid, sulfate, ammonium, black carbon, and organic carbon from ATom aircraft observations (black, ATom-1: a-e; ATom-2: f-j) and GEOS-Chem simulations by GC12 (blue), L2019

---

## Author Comment (AC3) · 24 Apr 2020

We thank the referee for the detailed reviews and constructive comments that help to improve the manuscript. Below we respond to the comments in detail. (Referee's comments are in Italic).

*This paper presented updated treatments of wet processes in GEOS-Chem, including rainout efficiencies for warm, mixed-phase and cold clouds, empirical washout by rain/snow, aqueous phase chemistry and wet removal for SO2 and sulfate, and wet surface uptakes during dry deposition. Model simulated concentrations of aerosols and aerosol precursors were evaluated with various surface observational data sets*

[Figure]

*over the U.S., Europe, Asia, and Arctic as well as aircraft measurements of nitric acid and aerosols during two ATom campaigns. Results showed significant improvement over previous version of the model and better agree with the observations. Although mentioned in various places in the paper, the roles of individual wet processes in the improvements were not systematically quantified. This paper is well organized and overall well written, but needs careful proofreading. I recommend publication after the following comments are addressed.*

We appreciate the positive comment about the paper. The revised manuscript has been carefully proofread.

*P4, Line 3, eqn 1: Pr is the grid-box large-scale precipitation (rain+snow) formation rate. LCW is liquid phase cloud water content. But the total condensed water content should also include ice cloud water content, which is missing from this equation.*

Yes, it is right. The equation and associated discussions have been modified.

*P8, Line 7, eqn 11: same issue as for eqn 1. For T>=258K (warm clouds), this equation assumes zero ice cloud water (ICW), which is probably not true in MERRA-2. Since the model uses temperature ranges to separate scavenging due to warm/mixedphase/cold clouds, the cloud condensed water (for all T) needs to include ICW. This is expected to have a significant impact on the model results of this paper.*

Thanks for pointing this out. We have modified the equation and code. We rerun the WETrev case with these updates.

*P10-11, Section 2.4: For rainout in cold cloud (T<237K), do you limit it to below the MERRA-2 tropopause?*

We did not. After discussed with the GEOS-Chem Steering Committee, we decided to limit rainout to below the MERRA-2 troposphere since stratospheric water in MERRA-2 is known to have unphysical behavior. We rerun the WETrev case with these updates. This has been clarified in the revised text.

*P16, L17-19, and Fig.5: Please double check. It looks like the L2019 and WETrev lines for HNO3 are switched. What aspect of the "old treatments in GC12" do you mean here?*

The lines of L2019 and WETrev cases in Fig.5 are right. GC12 and L2019 only considered washout of nitric acid by rain. WETrev also considered washouts of nitric acid by snow and ice which were absent in L2019 and GC12, therefore nitric acid concentrations of WETrev between 500 hPa and 300 hPa are lower than those of L2019. Rainout efficiency of nitric acid by cold cloud in WETrev is lower than that of L2019, therefore, nitric acid concentrations of WETrev above 300 hPa are higher than those of L2019. Old treatment referred to cold cloud wet scavenging of nitric acid in GC12 is treated the same as water-soluble aerosol with 100 % rainout efficiency. Cold cloud rainout efficiency in WETrev is based on the parameterization of nitric acid partitioning in cold cloud developed by Kärcher et al. (2008). We modified the sentence as: As we mentioned earlier, L2019 may overestimate cold cloud wet scavenging of nitric acid due to treat cold cloud rainout of nitric acid the same as water-soluble aerosol with 100 % rainout efficiency.

*P19, Code and data availability: the revised GEOS-Chem v12.6.0 code and model output need to be made available at a public data depository. Also it's not clear where the various observational data sets used in this work were downloaded from.*

We have updated the Code and data availability. Links of observational data sets have been provided.

*Minor comments: P4, Line 28: is LW different than LCW in eqn 1.*

LW in equation 1 is liquid water content for Henry's law. It equals liquid cloud water content (LCW) in the atmosphere.

*P5, L1 (and other places): Do you mean "acidity"?*

Yes, you are right. Modified.

*P5, L3: H\* can be calculated . . .*

Modified.

*P5, L8-9: what are the units for these constants and coefficients?*

We modified the sentence as: where HSO2, HH2O2, and HNH3 are the Henry's law constants (M atm-1) for SO2, H2O2, and NH3, respectively. K1 (M), K2 (M), K3 (M), K4 (M2), and K5 (M) are rate coefficients for SO2 reaction, HSO-3 reaction, H2O2 reaction, H2O reaction, and NH3 reaction, respectively.

*P6, L19: the comma is misplaced.*

Modified.

*P7, L20: LCW not LWC*

Modified.

*P13, L3: Emissions are produced by the default setting of HEMCO. Does this mean that emissions are specific to the periods of ATom-1 and ATom-2 campaigns?*

We used the default setting of HEMCO to produce emissions for all simulations presented in this work. We modified the sentence as: Emission over Europe is produced by EMEP inventory. Other emissions are produced by the default setting of HEMCO (Keller et al., 2014) for all simulations presented in this work. EMEP emission over Europe is used in our rerun cases of GC12, L2019, and WETrev. It is because we found the replacement of EMEP emission with CEDS global emission in GC12.6.0 leads unreasonable performance of ammonia seasonal variation over Europe.

*P13, L16: Is there a reference for "a large amount of USEPA observations are located at urban regions"?*

We didn't find related reference. There were 288 EPA's Air Quality System sites with valid data in each month of 2011. Only 69 of these sites were with the mark of 'Not in

a city'. More information can be found at https://www.epa.gov/outdoor-air-quality-data.

*P14, L20: remove "– ".*

Modified.

*P15, L7: low dissolution*

Modified.

*P15, L27  L29: "at Alert during spring" – during winter / early spring?*

For BC at Alert, it is winter and spring. For sulfate at Alert, it is spring. We have modified the sentence.

*P16, L1: converted*

Modified.

*P16, L10-11: Why are the flight tracks over the land filtered out for comparison?*

ATom observations over the land, whose values vary greatly, only account for 28 % of total measurements. To make the comparison more appropriate, we filtered out the flight tracks over the land.

*P18, L18-24: this sentence needs a break.*

We rewrote the sentence as: In this study, we updated aqueous phase chemistry and wet scavenging for SO2 and sulfate, rainout efficiencies for warm, mixed, and cold cloud, empirical washout by rain and snow, and wet surface uptakes during dry deposition in GEOS-Chem version 12.6.0. Systematic validations of simulated aerosol precursors and aerosols with ground based monitoring networks over the US, Europe, and Asia, in-site observations at Arctic for surface mass concentrations and aircraft measurements during ATom-1 and ATom2 for their vertical profiles were presented.

*P19, L3, L12: remove"an"; exist.*

Modified.

*Table 1: refer the reader to eqn 16.*

Accepted.

---

## Author Response (AR2)

Dear Samuel,

Thank you very much for handling our paper. Jose Jimenez's group contacted with us for the introduction of ATom observations in our paper. After a detailed discussion with them, we would like to add some background information of ATom observations in the final paper:

1. the data cite these DOIs for ATom-1 and 2 AMS and for whole dataset:
https://doi.org/10.3334/ORNLDAAC/1716
https://doi.org/10.3334/ORNLDAAC/1581

2. Instruments for nitric acid and aerosols measurement during ATom:
Nitric acid was measured by Chemical Ionization Mass Spectrometer, while aerosols were measured by CU Aircraft High-Resolution Time-of-Flight Aerosol Mass Spectrometer (Hodzic et al., 2020). The work of Brock et al. (2019) indicated that there is very good quantitative agreement between AMS and volume data.

3. Explanation of OC comparison in this work:
For ATom data, OC is calculated by OA_PM1_AMS/OAtoOC_PM1_AMS. For model, we used 1.8 for SOAs.

4. the data source (https://daac.ornl.gov/ATOM/campaign/) was acknowledged in the paper.

These changes were added in the final paper. Let us know if you have any questions.

Best,

Gan